# The aberrant upregulation of exon 10-inclusive SREK1 through SRSF10 acts as an oncogenic driver in human hepatocellular carcinoma

Cunjie Chang [1,2,14], Muthukumar Rajasekaran[3,14], Yiting Qiao[4,14], Heng Dong[1,14], Yu Wang [3], Hongping Xia [3], Amudha Deivasigamani[3], Minjie Wu[1], Karthik Sekar[3], Hengjun Gao[5], Mengqing Sun[1], Yuqin Niu[6], Qian Li[1], Lin Tao[1,2], Zhen Yan[1], Menglan Wang[1], Shasha Chen[7], Shujuan Zhao[1,2], Dajing Chen[1,2], Lina Li[1], Fan Yang[1], Haojin Gao[1], Baodong Chen[1], Ling Su[1], Liang Xu [8], Ye Chen [8], Veerabrahma Pratap Seshachalam[3], Gongxing Chen[1,2], Jayantha Gunaratne[9], Wanjin Hong [9], Junping Shi[10], Gongying Chen[10], David S. Grierson[11], Benoit Chabot [12], Tian Xie [1,2✉], Kam Man Hui [1,3,9,13✉] & Jianxiang Chen [1,2,3✉]

Deregulation of alternative splicing is implicated as a relevant source of molecular heterogeneity in cancer. However, the targets and intrinsic mechanisms of splicing in hepatocarcinogenesis are largely unknown. Here, we report a functional impact of a Splicing Regulatory Glutamine/Lysine-Rich Protein 1 (SREK1) variant and its regulator, Serine/arginine-rich splicing factor 10 (SRSF10). HCC patients with poor prognosis express higher levels of exon 10-inclusive SREK1 (SREK1$^L$). SREK1$^L$ can sustain BLOC1S5-TXNDC5 (B-T) expression, a targeted gene of nonsense-mediated mRNA decay through inhibiting exon-exon junction complex binding with B-T to exert its oncogenic role. B-T plays its competing endogenous RNA role by inhibiting miR-30c-5p and miR-30e-5p, and further promoting the expression of downstream oncogenic targets SRSF10 and TXNDC5. Interestingly, SRSF10 can act as a splicing regulator for SREK1$^L$ to promote hepatocarcinogenesis via the formation of a SRSF10-associated complex. In summary, we demonstrate a SRSF10/SREK1$^L$/B-T signalling loop to accelerate the hepatocarcinogenesis.

[1] School of Pharmacy and Department of Hepatology, the Affiliated Hospital of Hangzhou Normal University, Hangzhou Normal University, Hangzhou 311121, P. R. China. [2] Key Laboratory of Elemene Class Anti-Cancer Chinese Medicines, Engineering Laboratory of Development and Application of Traditional Chinese Medicines, Collaborative Innovation Center of Traditional Chinese Medicines of Zhejiang Province, School of Pharmacy, Hangzhou Normal University, Hangzhou 311121, P. R. China. [3] Laboratory of Cancer Genomics, Division of Cellular and Molecular Research, National Cancer Centre, Singapore 169610, Singapore. [4] The First Affiliated Hospital, Key Laboratory of Combined Multi-Organ Transplantation, Ministry of Public Health, Key Laboratory of Organ Transplantation of Zhejiang Province, School of Medicine, Zhejiang University, Hangzhou 310003, P. R. China. [5] Department of Hepatopancreatobiliary Surgery, Shandong Provincial Hospital, Cheeloo College of Medicine, Shandong University, Ji'nan, Shandong 250021, P.R. China. [6] The First Affiliated Hospital, School of Medicine, Shihezi University, Shihezi 832008, P. R. China. [7] Department of Traditional Chinese Medicine, Taizhou Cancer Hospital, Wenling 317502, China. [8] Cancer Science Institute of Singapore, National University of Singapore, Singapore 117599, Singapore. [9] Institute of Molecular and Cell Biology, A*STAR, Biopolis Drive Proteos, Singapore 138673, Singapore. [10] Institute of Hepatology and Metabolic Diseases, Department of Hepatology, the Affiliated Hospital of Hangzhou Normal University, Hangzhou 311121, China. [11] Faculty of Pharmaceutical Sciences, University of British Columbia, Vancouver, BC V6T 1Z3, Canada. [12] Département de Microbiologie et d'Infectiologie, Faculté de Médecine et des Sciences de la Santé, Université de Sherbrooke, Sherbrooke, QC J1E 4K8, Canada. [13] Program in Cancer and Stem Cell Biology, Duke-NUS Medical School, Singapore 169857, Singapore. [14] These authors contributed equally: Cunjie Chang, Muthukumar Rajasekaran, Yiting Qiao, Heng Dong. ✉email: xbs@hznu.edu.cn; cmrhkm@nccs.com.sg; chenjx@hznu.edu.cn

Liver cancer is a major global health problem whose incidence is on the rise[1]. Hepatocellular carcinoma (HCC) is the third leading cause of cancer mortality worldwide[2]. HCC is a molecularly heterogeneous tumor and this heterogeneity can decrease the efficacy of targeted therapy[3]. Recently, the deregulation of alternative splicing (AS) regulators has been implicated in generating this tumor heterogeneity[4]. Therefore, a thorough compilation of alterations in the splicing machinery of HCC cells may help identify key biomarkers and promising therapeutic strategies.

AS is a precisely regulated posttranscriptional process that controls gene expression and generates proteomic diversity[5,6]. The deregulation of isoforms of splicing regulators and/or corresponding splicing regulators has been frequently observed in human disease, notably in cancers, including HCC[7–9]. The intrinsic balance between functional isoforms is essential for the physiological functions of cancer cells and their ability to maintain proliferative potential[8,9]. Characterizing the functional impact of AS variants, their regulators, and the signaling pathways involved is therefore critical for interpreting the effects of aberrant isoform expression leading to hepatocarcinogenesis and/or metastasis, and for the rational design of therapeutic strategies.

Some of the AS factors could be further modulated by the inclusion or exclusion of their specific exon(s) coded for the domain(s) with crucial regulatory roles[10–12], which can be observed in human cancer. One of the notable reported AS factors is the SREK1 (also named SRrp86 or SFRS12), a serine/arginine-rich (SR) splicing protein containing an unusual glutamic acid-lysine (EK)-rich domain[13], which has been reported to modulate SR-rich protein activity and splice site selection by the AS of its exon 10-coding EK domain[14,15]. The function of SREK1 in cancer is largely unknown. Serine/arginine-rich splicing factor (SRSF)10 (also named SRp38 or FUSIP1) is a member of the SR family of proteins, which are involved in RNA splicing by their phosphorylation and interactions with small nuclear ribonucleoprotein particles[16,17]. Recently, SRSF10 has been implicated in the DNA damage response[18,19], HIV replication[20], differentiation and glucose production[21], and colon and cervical cancer progression[22,23].

In this work, we find the high expression of an exon 10-inclusive form of SREK1 (SREK1$^L$) in HCC tumor tissues (HCC-T) that is associated with poor prognosis of the patients, and demonstrate that SREK1$^L$ promotes the oncogenesis of HCC cells in vitro and in vivo through its interactions with NMD components to regulate the expression of BLOC1S5-TXNDC5 (B-T). B-T is verified as a competing endogenous RNA (ceRNA) for miR-30c/e-5p to maintain the high expression of TXNDC5 and SRSF10. SRSF10 is shown to be significantly upregulated in HCC-T and further promote the oncogenesis of HCC cells by maintaining the inclusion of SREK1 exon 10.

## Results

**SREK1$^L$ is enriched in HCC and associated with prognosis in HCC patients**. Human SREK1 has 13 exons, among which alternative exon 10, which codes for a glutamic acid-lysine (EK)-rich domain, is subject to AS regulation established before[13]. Exon 10 inclusion or exclusion generates two SREK1 transcripts, SREK1$^L$ and SREK1$^S$, respectively (Supplementary Fig. 1a). To detect the exon 10-inclusive and -exclusive forms, three sets of primers were designed (Supplementary Fig. 1b). SREK1$^L$ and SREK1$^S$ were quantified with primer set 1, which covered a segment spanning exon 9 to exon 11. Moreover, primer set 2, which covered the exon 9/exon 10 and exon 10/exon 11 junctions, was designed to further validate SREK1$^L$. Primer set 3 covered the exon 9/exon 11 junction and was used to detect SREK1$^S$

(Supplementary Fig. 1b). To explore the role of SREK1 splicing in HCC, we validated the splicing using a tissue set comprising 10 pairs of HCC tissues (Supplementary Table 1). The expression of SREK1$^L$ and SREK1$^S$ was determined by PCR using primer set 1 (Supplementary Fig. 1b), and the percentage-splice-in (PSI) was analyzed in each sample, which was significantly higher in HCC-T than in matched normal tissues (HCC-MN) (Fig. 1a, b). The result was further validated by a tissue set consisting of 60 pairs of HCC tissues (Supplementary Table 2). The expression of SREK1$^L$ and SREK1$^S$ was evaluated by determining the expression copy numbers of two variants normalized to the copy numbers of the corresponding Flag-SREK1$^L$ and Flag-SREK1$^S$ plasmid DNA, respectively. These results confirmed that the expression of SREK1$^L$ and, to a lesser extent, SREK1$^S$ was significantly higher in HCC-T than in HCC-MN (Fig. 1c and Supplementary Fig. 1c). Moreover, the increased expression of SREK1$^L$ was significantly correlated with poorer overall survival (OS) and disease-free survival (DFS) in HCC patients (Fig. 1d). This was not observed with SREK1$^S$ (Supplementary Fig. 1d). To evaluate the relative splicing of the two variants, the SREK1$^L$/SREK1$^S$ expression ratio was determined, which was significantly higher in HCC-T than in HCC-MN (Fig. 1e). A high SREK1$^L$/SREK1$^S$ expression ratio was also significantly correlated with poorer OS and DFS (Fig. 1f). To investigate the protein expression of SREK1$^L$, we created an antibody targeting SREK1$^L$ raised against a peptide encoded in exon 10 (CRSKEIDEKRKKDKK), and IHC staining was employed to study the expression and localization of SREK1$^L$ in tissues. SREK1$^L$ showed strong nuclear and weak cytoplasmic staining in HCC tissues (Fig. 1g), and its expression was significantly higher in HCC-T than in HCC-MN (Fig. 1g and Supplementary Fig. 1e). To study the potential prognostic significance of SREK1$^L$ protein level, the expression of SREK1$^L$ in 48 HCC samples was evaluated by IHC staining (Supplementary Table 3). High SREK1$^L$ expression was significantly correlated with poorer OS and DFS in HCC patients (Fig. 1h).

Our data indicate that the alternatively spliced variant SREK1$^L$ is enriched in HCC-T and that its increased expression significantly correlates with poorer prognosis in HCC patients, indicating that SREK1$^L$ expression may functionally contribute to hepatocarcinogenesis.

**SREK1$^L$ promotes the hepatocarcinogenesis**. To explore the function of SREK1$^L$ in HCC, two siRNAs or shRNAs targeting exon 10 were designed to transiently or stably silence SREK1$^L$ expression, respectively, in Hep3B and HCCLM3 cells (Supplementary Fig. 2a, b). Knockdown of SREK1$^L$ inhibited the growth (Fig. 2a and Supplementary Fig. 2c) and delayed the migration in Hep3B and HCCLM3 cells (Supplementary Fig. 2d, e). Moreover, anchorage-independent soft agar colony formation assays indicated that the knockdown of SREK1$^L$ markedly reduced the colony numbers of Hep3B and HCCLM3 cells compared to those of control scramble (Scram)-transfected cells (Fig. 2b and Supplementary Fig. 2f). To evaluate the role of SREK1$^S$ variant in cell growth, we employed and tested the knockdown efficiency of three siRNAs targeting exon 9/exon 11 junction sequence in HCC cells (Supplementary Fig. 2g). Our data indicated that depletion of SREK1$^S$ variant specifically in Hep3B cells has no effect on the cell growth (Supplementary Fig. 2h, i); thus, we focused on the functional study of SREK1$^L$ further. SREK1$^L$ was highly expressed in the embryonic liver (with a peak at approximately E14.5) but markedly inhibited in the adult liver (Supplementary Fig. 2j) by a reported database[24], indicating its potential involvement in embryonic liver stem cells. Here, we employed oncosphere assays to investigate whether SREK1$^L$ has a role in the cancer stem cells (CSCs) that is crucial for

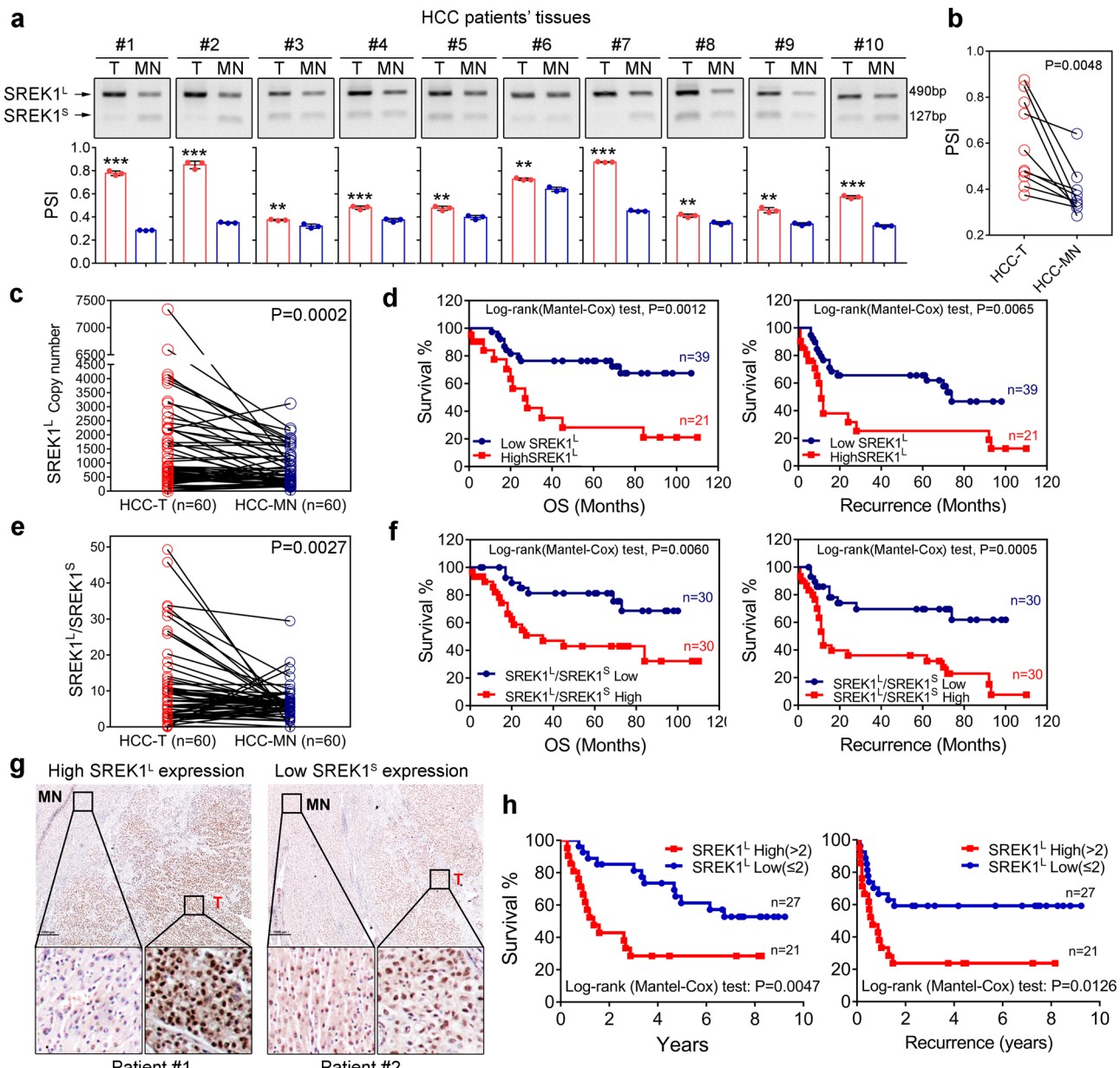

**Fig. 1 The variant SREK1$^L$ is enriched in HCC and associated with prognosis.** PCR detection with primer set 1 and quantification of two variants in ten pairs of HCC patient tissues (Supplementary Table 1). Quantified relative PSI (Percentage-splice-in = splice_in/(splice_in+splice_out)) of the variants **a** in each tissues of ten pairs' samples ($n = 3$, data are shown as the mean ± SD, two-tailed, unpaired $t$ test is used) or **b** the average PSI (Percentage-splice-in = splice_in/(splice_in+splice_out)) in ten pairs tissues are shown (red column indicates tumor (T), Blue column indicates matched normal tissues (MN)), $n = 10$. **c** Normalized SREK1$^L$ copy number in 60 pairs of HCC tissues (Supplementary Table 2) and **d** survival analysis of 60 HCC patients with high and low expression of SREK1$^L$ (using the mean as the cut-off value; OS overall survival). **e** Normalized SREK1$^L$/SREK1$^S$ expression ratios in 60 pairs of HCC tissues (Supplementary Table 2), and **f** survival analysis of 60 HCC patients with high or low SREK1$^L$/SREK1$^S$ expression (using the median as the cut-off value). **g** Representative immunohistochemical (IHC) staining figures of high or low SREK1$^L$ expression in HCC tissues with SREK1$^L$ detection antibody, scale bar = 1000 μm. **h** Survival analysis on high or low SREK1$^L$ expression by IHC in 48 HCC patients was determined and scored independently by three pathologists. Two-tailed, paired $t$ test is used for (**b**, **c**, **e**), **$p < 0.01$; ***$p < 0.001$. Source data are provided as a Source Data file.

hepatocarcinogenesis[25]. Oncosphere assays indicated that the stable knockdown of SREK1$^L$ markedly inhibited the construction and numbers of oncospheres when compared to those in Scram control cells (Fig. 2c), indicating a role of SREK1$^L$ in promoting HCC CSC. To further investigate the oncogenic role of SREK1$^L$ in vivo, a subcutaneous xenograft model of human Hep3B cells in which Scram or SREK1$^L$ had been stably knocked down showed a remarkable delay in tumor growth compared to that in stable Scram-transfected cells (Fig. 2d, e). To further verify the role of SREK1$^L$ in hepatocyte proliferation, we forced

mouse SREK1$^L$ and GFP control expression in mouse liver by adenovirus-mediated gene delivery for 40 days (Supplementary Fig. 2k, l), and further performed surgery by removing a part of the liver to monitor the proliferation of the hepatocytes. It was found that expression of SREK1$^L$ compared with GFP control could significantly promote the cell division rate (labeled by phosphorylated Histone 3 Ser10, pH3S10) to accelerate the proliferation of the hepatocyte during the regeneration process (Supplementary Fig. 2m, n). Our data suggest that SREK1$^L$ represents a key oncogenic driver of HCC.

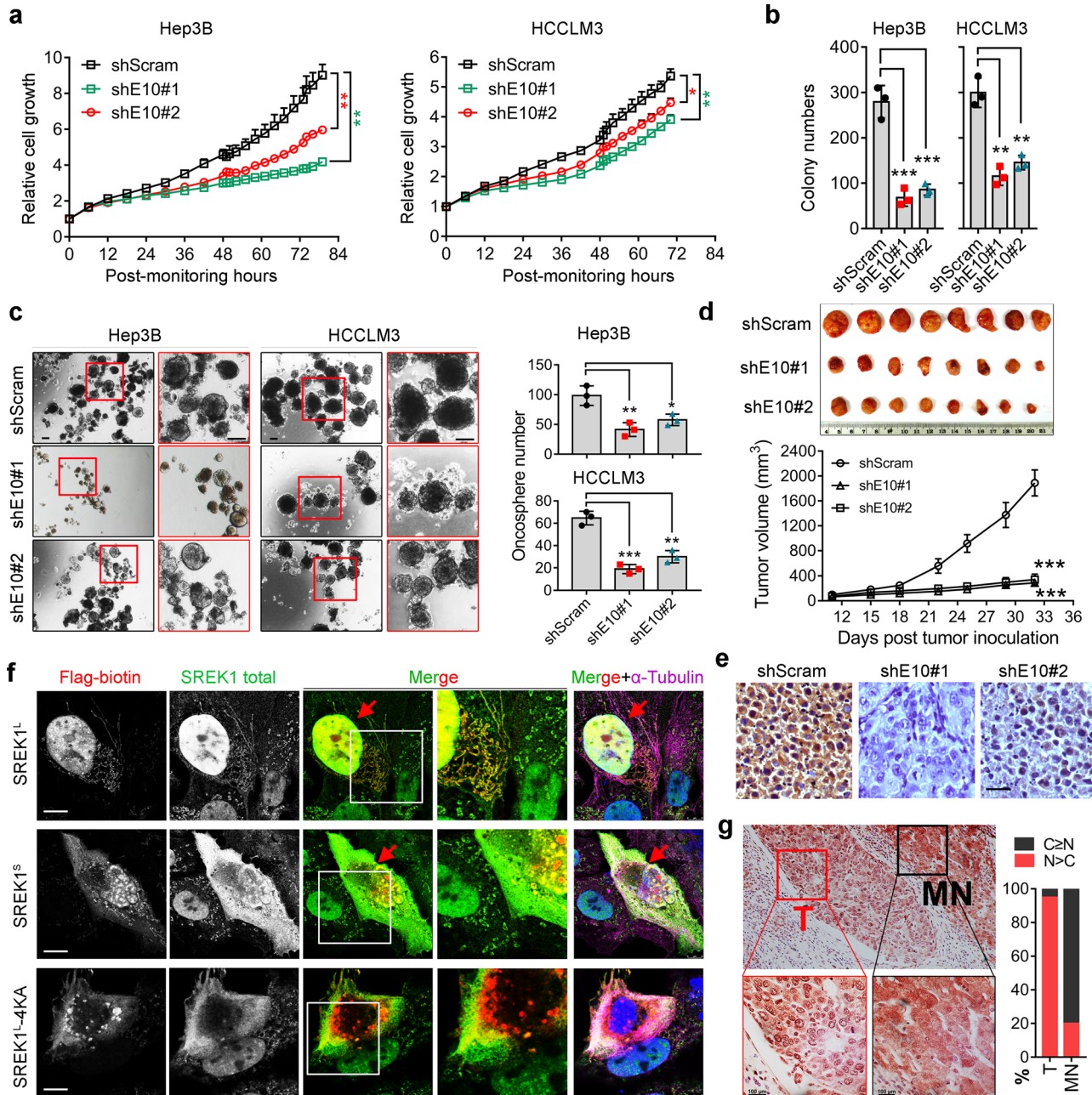

**Fig. 2 SREK1$^L$ is an oncogenic driver with nuclear localization in HCC. a** Cell growth analysis ($n = 3$), data are shown as the mean ± SEM. **b** Anchorage-independent soft agar colony formation assays ($n = 3$), and **c** oncosphere assays ($n = 3$) of cells in which scramble control or exon 10-specific SREK1 was stably knocked down (scale bar = 200 μm), data are shown as the mean ± SD. **d** Mouse tumorigenic assays of Hep3B cells with scramble control or exon 10-specific SREK1 stably knocked down ($n = 8$, data are shown as the mean ± SEM). **e** The immunohistochemical staining of the representative xenograft tumor tissues, scale bar = 100 μm. **f** Confocal microscopy showing immunofluorescence staining for Flag-tagged SREK1$^L$ and SREK1$^S$ (shown in red) and their colocalization with endogenous SREK1 (shown in green), microtubules (shown in purple) and nuclei (stained with DAPI, shown in blue) in HCCLM3 cells. Scale bar = 5 μm. **g** Representative immunohistochemical staining of HCC patient tissues for endogenous SREK1 (Scale bar = 100 μm, left panel); T tumor tissues, MN matched normal tissues, C cytoplasm, N nucleus. The amount of SREK1 in the cytoplasm or nuclei of cells in tumor or matched normal tissues from HCC patients was quantified ($n = 10$, right panel), C>N indicates more specific SREK1 detected in the cytoplasm, N>C indicates more specific SREK1 staining in the nucleus. Two-tailed, unpaired $t$ test is used for statistical analysis of (**a, b, c, e**), *$p < 0.05$; **$p < 0.01$; ***$p < 0.001$. Source data are provided as a Source Data file.

SREK1$^L$ has been detected with nuclear localization in HCC tissues (Fig. 1h), and a putative consensus sequence of the bipartite nuclear localization signal (NLS)–KKDKK[26] has been identified in EK domain (Supplementary Fig. 2o). We therefore speculate whether the exon 10 inclusion alters the subcellular localization of SREK1 via selective expression of the EK domain. Flag-SREK1$^L$ and Flag-SREK1$^S$ constructs were transfected into

HCCLM3 cells. Antibodies against Flag, total SREK1 (detecting both forms of SREK1) and α-Tubulin were used to stain and assess their localization in HCC cells (Fig. 2f). Confocal microscopy confirmed that endogenous SREK1 localized mainly to the nucleus (Fig. 2f). Flag-SREK1$^L$ was mainly detected in the nucleus, while Flag-SREK1$^S$ and an NLS mutant named Flag-SREK1$^L$-4KA generated by the mutation of all four Lysine (K)

into Alanine (A) showed a whole-cell distribution and much less nuclear localization than Flag-SREK1$^L$ (Fig. 2f and Supplementary Fig. 2o). This confirms that the EK domain contributes to nuclear localization of SREK1 in HCC cells. As more SREK1$^L$ was detected in HCC tumor than in HCC-MN tissues (Fig. 1), we detected SREK1 expression in slides of HCC-MN tissues using an antibody recognizing both forms. Interestingly, SREK1 showed strong nuclear localization in HCC-T but weak cytoplasmic and nuclear localization in MN tissues (Fig. 2g), suggesting a role for nuclear SREK1$^L$ in hepatocarcinogenesis.

**SREK1$^L$ is involved in NMD signaling to promote B-T expression and HCC cell growth.** To explore the regulatory impact of SREK1$^L$, we specifically silenced SREK1$^L$ expression in Hep3B and HCCLM3 cells (Supplementary Fig. 2a) followed by transcriptome sequencing. SREK1 has previously been suggested to play a role in splicing regulation[14,15]; therefore, we analyzed the 5 types of AS to identify potential SREK1$^L$-controlled splicing events common to both Hep3B and HCCLM3 cells (Supplementary Fig. 3a) by rMATS[27]. Surprisingly, no common SREK1$^L$ splicing events in two cell lines were found (Supplementary Fig. 3a), which might be due to its cell line-dependent splicing regulation, the expression differences of SREK1$^L$, SREK1$^S$ and other variants, or could indicate that SREK1 exerts its oncogenic role possibly via a mechanism other than AS. We further analyzed SREK1$^L$-controlled genes in our transcriptome sequencing data from Hep3B and HCCLM3 cells by a cut-off $P < 0.05$ and $log_2|FC|>2$ (Fig. 3a). Interestingly, three of five identified common SREK1$^L$-controlled-genes, including ABHD14A-ACY1 (A-A), SYS1-DBNDD2 (S-D) and B-T, are all NMD targeted genes (Fig. 3a). Furthermore, through the Kyoto Encyclopedia of Genes and Genomes analysis with the top 100 SREK1-correlated genes in TCGA HCC database, SREK1 expression was found significantly correlated with mRNA surveillance pathway mainly controlled by NMD pathway (Supplementary Fig. 3b). To investigate the correlation of SREK1$^L$ expression with three NMD candidate genes, we evaluated the expression of these genes in 24 HCC cell lines and in 60 pairs of HCC tissues, and found that endogenous expression of B-T was much higher than A-A and S-D expression in HCC cell lines (Supplementary Fig. 3c). Moreover, only the expression of B-T was correlated well with SREK1$^L$ expression both in the HCC cell lines and tissues (Fig. 3b). B-T and S-D expression, but not A-A expression, was significantly upregulated in HCC-T compared to HCC-MN tissues (Fig. 3c), and only the expression of B-T was significantly correlated with the OS and recurrence of HCC patients (Fig. 3d and Supplementary Fig. 3d). The expression of B-T was also significantly correlated with the OS of HCC patients in TCGA HCC database by GEPIA analysis[28] (Supplementary Fig. 3e). Moreover, B-T expression was attenuated markedly by silencing SREK1$^L$, but not SREK1$^S$ expression in Hep3B and HCCLM3 cells (Supplementary Fig. 3f). Our data confirmed that SREK1$^L$ is required for the regulation of B-T expression that is significantly correlated with HCC prognosis.

In mammals, the production of NMD targeted genes can be regulated by SMG1-UPF1-eRF1-eRF3 (SURF) complex (Supplementary Fig. 3g) and enhanced by the exon-exon junction complex (EJC)[29]. To investigate whether SREK1$^L$ is directly involved in these complexes, endogenous immunoprecipitation (IP) was performed using two anti-SREK1 antibodies in Hep3B cells. IP analysis showed that SREK1$^L$ was associated with two factors of SURF complex including UPF1 and MOV10, and with more factors of the EJC including UPF2, UPF3, MAGOH, GNL2 and SEC13 (Fig. 3e), indicating that SREK1$^L$ is indeed involved in these complexes, especially with the EJC. To further confirm

whether SREK1 could act as an EJC component via binding to B-T to control its expression, an RNA immunoprecipitation (RIP) assay was performed by SREK1 antibodies and we found that SREK1 indeed could directly bind to B-T transcript (Fig. 3f). To further investigate whether SREK1$^L$ could regulate the association of the EJC and SURF with B-T transcript, the RIP assay by UPF1, MOV10, UPF3, MAGOH and IgG control antibodies was performed in the scram control or SREK1$^L$ knockdown Hep3B cells (Fig. 3g and Supplementary Fig. 3h). We found that SREK1$^L$ knockdown could not affect the association of SURF components UPF1 and MOV10 with B-T transcript; however, the knockdown could markedly enhance the association of EJC components UPF3 and MAGOH with the B-T transcript (Fig. 3g). To further confirm it, we investigate the SREK1$^L$ and EJC deposition on the B-T or B-T ΔBS (SREK1$^L$ binding sites deleted) mRNA in Hep3B nuclear lysate, and found that increasing SREK1$^L$ expression could inhibit the association of two EJC components UPF2 and MAGOH with the B-T, but not B-T ΔBS RNA (Supplementary Fig. 3i). This indicates that the nuclear SREK1$^L$ binds with B-T transcript and protects it from EJC deposition to inhibit the subsequent NMD signaling and further sustain the expression of B-T in HCC cells (Fig. 3h).

As the expression noncoding RNA B-T is high endogenously and potentially sustained by SREK1$^L$ in HCC cells, we speculated that B-T might be important and involved in the cancer promotion role of SREK1$^L$. To verify it, we silenced the endogenous expression of B-T in Hep3B cells or re-expressed B-T in SREK1$^L$ knockdown cells (Supplementary Fig. 3j). B-T and SREK1$^L$ knockdown could delay cell growth, proliferation and colony numbers (Supplementary Fig. 3j, k). Moreover, the re-expression of B-T in SREK1$^L$ knockdown cells could significantly reverse the inhibition effect on cell growth, proliferation and colony formation by SREK1$^L$ knockdown (Supplementary Fig. 3j, k). To further verify whether SREK1$^L$ functions via regulating NMD target, we silenced UPF1, a critical NMD regulator and further evaluated the SREK1$^L$ proliferative role in HCCLM3 and Hep3B (Supplementary Fig. 3l). We found that SREK1$^L$ knockdown failed to inhibit the cell proliferation when UPF1 was depleted (Supplementary Fig. 3l), confirming the NMD regulation is important for SREK1$^L$ function in HCC cells.These data confirmed that B-T acts as a functional effector of SREK1$^L$ to promote HCC.

**B-T acts as a ceRNA for miR-30c-5p and miR-30e-5p to maintain the expression of TXNDC5 and SRSF10 and promote HCC.** To further explore the downstream regulation of B-T, we analyzed the potential ceRNA role of B-T by StarBase v2.0[30], and a most significant targeted gene—TXNDC5—was revealed (Fig. 4a). To identify the potential effect of B-T on the expression of BLOC1S5 and TXNDC5, two co-localized genes with B-T on chromosome 6, the endogenous expression of B-T was silenced or re-expression of B-T was performed transiently, and the RNA and protein expression of BLOC1S5 and TXNDC5 was analyzed in Hep3B cells (Fig. 4b and Supplementary Fig. 4a). Our data confirmed that knockdown or re-expression of B-T could inhibit or promote the expression of TXNDC5, but not BLOC1S5 (Fig. 4b and Supplementary Fig. 4a). To further explore the potential putative microRNAs binding both with B-T and 3'-untranslated region (3'-UTR) of TXNDC5 and also down-regulated in HCC-T, a Venn diagram analysis was performed and the miR-30c-5p and miR-30e-5p were generated finally (Fig. 4c).

We further verified that knockdown or re-expression of B-T could promote or inhibit the expression of miR-30c-5p and miR-30e-5p in Hep3B cells (Fig. 4d). To identify other common targeted genes by miR-30c-5p and miR-30e-5p that are also

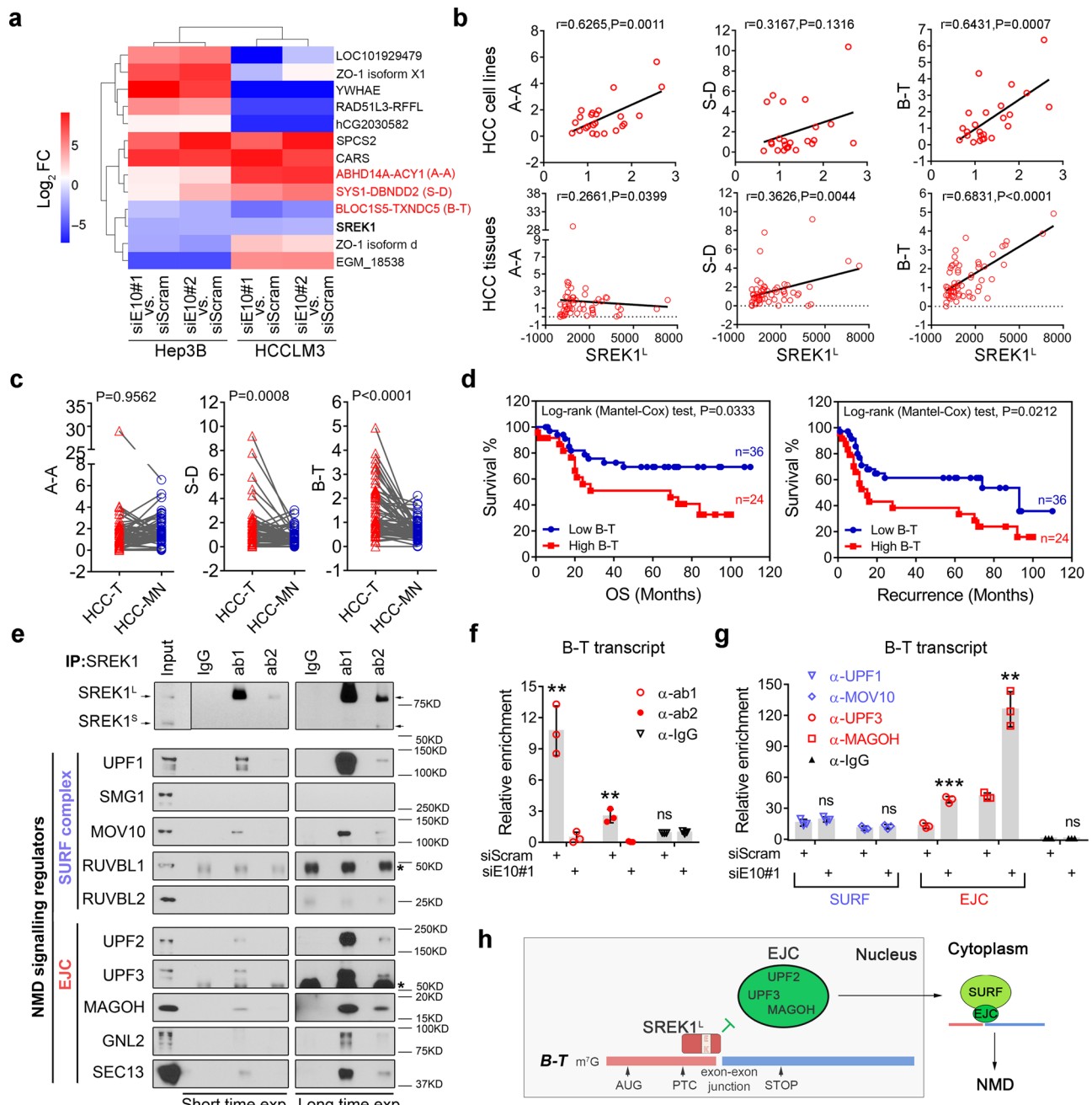

**Fig. 3 SREK1$^L$ maintains the expression of B-T, an NMD target, via inhibiting the exon-exon junction complex binding to the B-T transcript.**
**a** Expression heat map of scramble (siScram)-transfected or SREK1$^L$-knockdown (siE10#1 or #2) HCC cell lines mediated by siRNA. Targeted NMD genes are labeled in red. **b** Pearson correlation assay to assess the expression of three NMD target genes and SREK1$^L$ in 24 HCC cell lines or 60 HCC tumor tissues. **c** The expression of three NMD target genes in 60 tumor (T) and matched normal (MN) tissues, two-tailed, paired $t$ test is used. **d** Analysis of OS and disease-free survival in 60 HCC patients with either high or low B-T expression. **e** Endogenous immunoprecipitation of SREK1-associated components from the SURF or EJC complex in Hep3B cells using IgG or two commercial antibodies against SREK1 (ab1: recognizing SREK1$^L$, ab2: recognizing both L and S forms of SREK1). The star symbols indicate the heavy chain. The RNA immunoprecipitation assay for the detection of the enrichment of **f** SREK1, or the components of NMD complexes **g** UPF1, MOV10, UPF3, MAGOH on the B-T transcript in scram control or SREK1$^L$ knockdown Hep3B cells ($n = 3$, biologically independent experiments), data are shown as the mean ± SD, **$p < 0.01$, ***$p < 0.001$. **h** A proposed potential regulatory model of SREK1$^L$ protecting B-T from NMD via inhibiting the EJC binding to B-T transcript in HCC cells. Two-tailed, unpaired $t$ test is used for (**b**, **f**, **g**). Source data are provided as a Source Data file.

correlated well with SREK1 expression in HCC tissues, the downstream targets analysis of two microRNAs was further performed by Targetscan and miRDB[31,32], and the correlation of the targets with SREK1 expression in TCGA HCC database was also analyzed. Finally, eight genes were enriched in which SRSF10 scored as a top 1 candidate gene in HCC (Fig. 4e). To

further confirm the regulation of two microRNAs on both TXNDC5 and SRSF10, the sequences of 3'-UTR and the mutations of predicted binding sites were cloned into the luciferase reporter (Fig. 4f) and the reporter activity was examined. Transfection of the inhibitor or mimics of miR-30c-5p could significantly promote or inhibit the reporter activity of

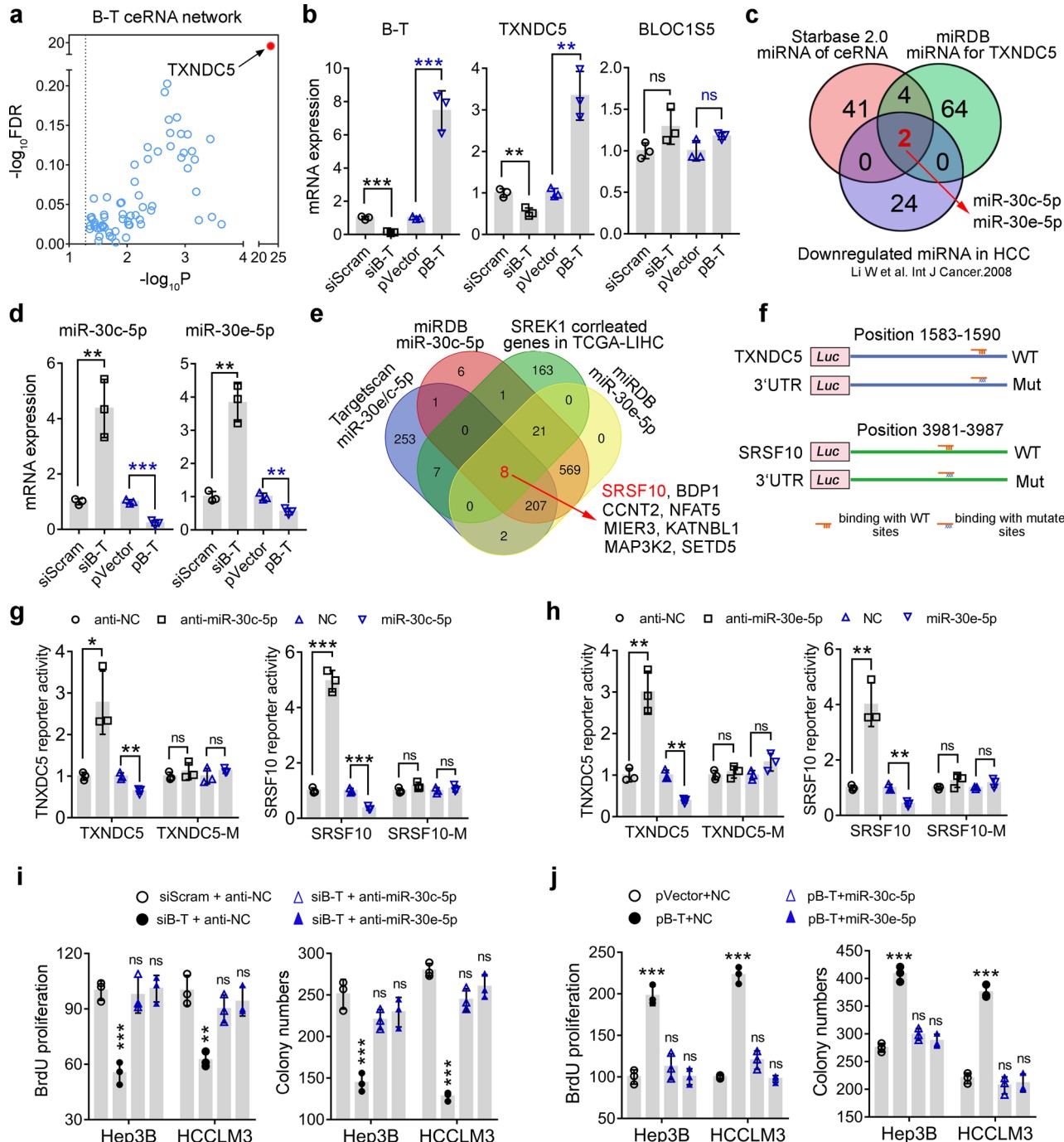

**Fig. 4 B-T plays its oncogenic role through acting as a ceRNA to inhibit miR-30c-5p and miR-30e-5p and to promote the expression of TXNDC5 and SRSF10 in HCC cells. a** The ceRNA plot analysis of B-T was shown by StarBase v2.0 based on the P and FDR value of the potential targets. **b** The real-time PCR analysis of the targeted genes after the knockdown or the forced expression of B-T in Hep3B cells. **c** Venn diagram analysis of the potential downstream microRNAs of B-T acting as a ceRNA in HCC. **d** The real-time PCR analysis of the effect of silenced or forced expression of B-T on targeted microRNAs' expression in Hep3B cells. **e** Venn diagram analysis of the common downstream targets of miR-30c-5p and miR-30e-5p correlated with SREK1 expression in HCC. The SRSF10 was identified and scored as the top 1 candidate. **f** The four 3'-UTR luciferase reporter constructs were presented with WT or mutated corresponding microRNAs binding sites. The effect of the inhibition or mimics expression of the **g** miR-30c-5p or **h** miR-30e-5p on WT or mutated corresponding 3'-UTR reporters' activity of TXDNC5 or SRSF10 in Hep3B cells. The BrdU proliferation or colony formation analysis of the Hep3B or HCCLM3 cells treated **i** by B-T knockdown or its combined knockdown with miR-30c-5p or miR-30e-5p, or **j** by forced B-T expression or the combined expression of B-T with miR-30c-5p or miR-30e-5p. $N = 3$, mean ± SD, two-tailed, unpaired $t$ test is used for (**b**, **d**, **g**–**j**), *$p < 0.05$; **$p < 0.01$; ***$p < 0.001$; ns indicates non-significant. Source data are provided as a Source Data file.

wildtype but not the mutated 3'-UTR of TXNDC5 and SRSF10 in Hep3B cells (Fig. 4g). A similar regulatory result was generated by miR-30e-5p (Fig. 4h). Moreover, transfection of the inhibitor or mimics of miR-30c-5p or miR-30e-5p could promote or inhibit the endogenous expression of both TXNDC5 and SRSF10 in Hep3B and HCCLM3 cells (Supplementary Fig. 4b). Thus, our data confirmed that B-T acts as a ceRNA for miR-30c-5p and miR-30e-5p inhibition that in turn maintain the expression of TXNDC5 and SRSF10, two reported oncogenic drivers in other cancers[22,23,33,34]. To further verify whether TXNDC5 mediated the role of SREK1 to promote cell growth, we re-expressed the TNXDC5 or control in SREK1$^L$ or scramble knockdown in HCCLM3 cells (Supplementary Fig. 4c) and found that re-expression of TNXDC5 could promote the cell growth rate and partially rescue the growth inhibition effect of SREK1$^L$ knockdown in HCCLM3 cells (Supplementary Fig. 4d).

To investigate whether miR-30c-5p and miR-30e-5p mediate the function of B-T, transfection of the inhibitors or mimics of these microRNAs together with the siRNA or overexpression clone of B-T was performed and the BrdU proliferation and colony formation assays were further evaluated in Hep3B and HCCLM3 cells. As mentioned above, knockdown of B-T could inhibit the proliferation and colony formation; while inhibiting two microRNAs together with B-T knockdown could completely reverse the inhibitory effect of B-T knockdown on the cell proliferation and colony formation in Hep3B and HCCLM3 cells (Fig. 4i). Furthermore, the expression of B-T could significantly promote the proliferation and colony formation, which could be inhibited markedly by the expression of two microRNAs mimics in Hep3B and HCCLM3 cells (Fig. 4j).

**SRSF10 sustains the exon 10 inclusion of SREK1 in HCC.** The above data indicate that exon 10 is required for SREK1 oncogenic activity via regulating B-T ceRNA network in HCC cells. To explore which splicing factors contribute to the inclusion, a motif analysis on exon 10 was performed to identify the conserved motifs for splicing factors using MEME (http://meme-suite.org/). The analysis found a conserved sequence highly enriched in the nucleotides G and A, consistent with the reported SRSF10-binding motif previously[23] (Fig. 5a). To confirm our observation, we silenced SRSF10 or SRSF1 (as a control) expression in HCCLM3, Huh1, SNU398 and Hep3B cells and the relative mRNA expression and expressed copies number of SREK1$^L$ and SREK1$^S$ were determined (Supplementary Fig. 5a and Fig. 5b). We found that SRSF1 knockdown had no significant effect on SREK1$^L$ or SREK1$^S$ expression; however, SRSF10 knockdown significantly attenuated SREK1$^L$ expression but increased SREK1$^S$ expression (Fig. 5b) compared to that in Scram knockdown cells. In stable knockdown cells, we also found that SRSF10 knockdown also significantly decreased SREK1$^L$ but increased SREK1$^S$ expression (Supplementary Fig. 5b). RNA-seq analysis of Scram-transfected and stable SRSF10-knockdown cells was performed, and the reads crossing exon 10 junctions were further analyzed using a Sashimi plot (Fig. 5c), indicating that SRSF10 knockdown markedly reduced the exon 10 reads but increased the reads across exons 9 and 11 (Fig. 5c), which confirmed that SRSF10 is essential for exon 10 inclusion. To determine whether SRSF10 binds directly to SREK1$^L$ mRNA, an SRSF10 IP antibody was employed (Supplementary Fig. 5c) to perform RIP assays in Hep3B and HCCLM3 cells (Fig. 5d). Four sets of primers designed from exon 10 were used to investigate the binding capabilities of SREK1$^L$ and BCLAF1 (a reported SRSF10 splicing target) pre-mRNAs with SRSF10 (Fig. 5d). We found that anti-SRSF10, but not IgG and anti-SRSF1, could recover SREK1$^L$ or BCLAF1 mRNA (Fig. 5d, e and Supplementary Fig. 5d).

As exon 10 contributes to SREK1$^L$ nuclear localization, we speculated that SRSF10 also regulates the nuclear localization of SREK1$^L$. To confirm this, the localization of endogenous SREK1, SRSF10 and α-Tubulin in Scram-transfected and SRSF1- or SRSF10-knockdown cells was studied by confocal microscopy (Fig. 5f and Supplementary Fig. 5e). The knockdown of SRSF10, but not SRSF1, completely inhibited nuclear SRSF10 levels and attenuated the nuclear SREK1 expression in both Hep3B and HCCLM3 cells (Fig. 5f and Supplementary Fig. 5e). The nuclear-cytosol extraction further found that knockdown of SRSF10 could deplete the nuclear SREK1$^L$ and increase the SREK1$^S$ accumulated in the cytoplasm of HCCLM3 cells (Supplementary Fig. 5f), confirming that SRSF10 can promote nuclear SREK1$^L$ by maintaining exon 10 inclusion of SREK1 in HCC cells.

Using a panel of 25 HCC cell lines, the protein expression of SRSF10, SREK1$^L$, SREK1$^S$, SRSF1 and α-Tubulin was determined by western blotting (Fig. 5g), following which their expression was quantified, and correlation studies were performed (Fig. 5h). Our data indicate that SRSF10 and SRSF1 were highly expressed in HCC cell lines and that the expression of SRSF10, but not that of SRSF1, was positively correlated with SREK1$^L$ expression (Fig. 5h and Supplementary Fig. 5g). Moreover, in the studied cohort of 60 paired HCC tissues, SRSF10 expression was significantly correlated with SREK1$^L$ expression in only HCC-T ($r = 0.7016$, $P < 0.0001$) but not in HCC-MN ($r = 0.1895$, $P = 0.1470$) (Fig. 5i and Supplementary Fig. 5h). SRSF10 expression was not correlated well with SREK1$^S$ expression in HCC-T ($r = 0.3305$, $P = 0.0099$) or HCC-MN ($r = 0.2301$, $P = 0.0769$) (Supplementary Fig. 5i). Moreover, the SRSF10 expression was also significantly correlated with SREK1 expression in TCGA-LIHC database (Supplementary Fig. 5j), and in six pairs HCC tissue samples we found that the long and short SREK1 proteins were significantly alternative spliced in T and MN tissue by evaluating the protein expression (Supplementary Fig. 5k), confirming that the upregulation of SRSF10 indeed promotes SREK1$^L$ expression in HCC.

**SRSF10 correlates with the prognosis of HCC patients and acts as an oncogenic driver in HCC.** SRSFs have been implicated in cancer progression. Here, we analyzed the expression of SRSF10 and its correlation with prognosis using our tissue microarray database as previously reported[35]. Our analysis indicated that SRSF10 were significantly upregulated in HCC-T compared with HCC-MN in our microarray database and 60 pair tissues cohort (Fig. 6a, b). The upregulation of SRSF10, SRSF1 and SRSF12 was markedly correlated with poor OS and recurrence in HCC patients (Fig. 6c, Supplementary Fig. 6a, and Supplementary Table 4). The SRSF10 protein was evaluated in HCC patient tissues by IHC analysis, which further indicated that SRSF10 was highly expressed in HCC-T with strong nuclear localization (Supplementary Fig. 6b). Furthermore, SRSF10 expression was also significantly correlated with the recurrence of HCC (Supplementary Fig. 6c).

As SRSF10 was found to be upregulated and correlated with poor prognosis, we speculated that SRSF10, which has not yet been well studied in HCC, might be a driver for hepatocarcinogenesis. To confirm this, SRSF10 expression was stably silenced in Hep3B and HCCLM3 cells (Supplementary Fig. 6d), and the BrdU assay was employed to monitor cell proliferation (Supplementary Fig. 6e). The results confirmed that SRSF10 knockdown significantly attenuated the proliferation of HCC cells compared to that of Scram-transfected cells (Supplementary Fig. 6e), and markedly reduced the numbers of Hep3B and HCCLM3 cell colonies (Fig. 6d). A wound-healing migration assay showed that SRSF10 knockdown decreased the wound-healing closure rate in Hep3B and HCCLM3 cells (Supplementary Fig. 6f). Moreover, an oncosphere assay was also performed

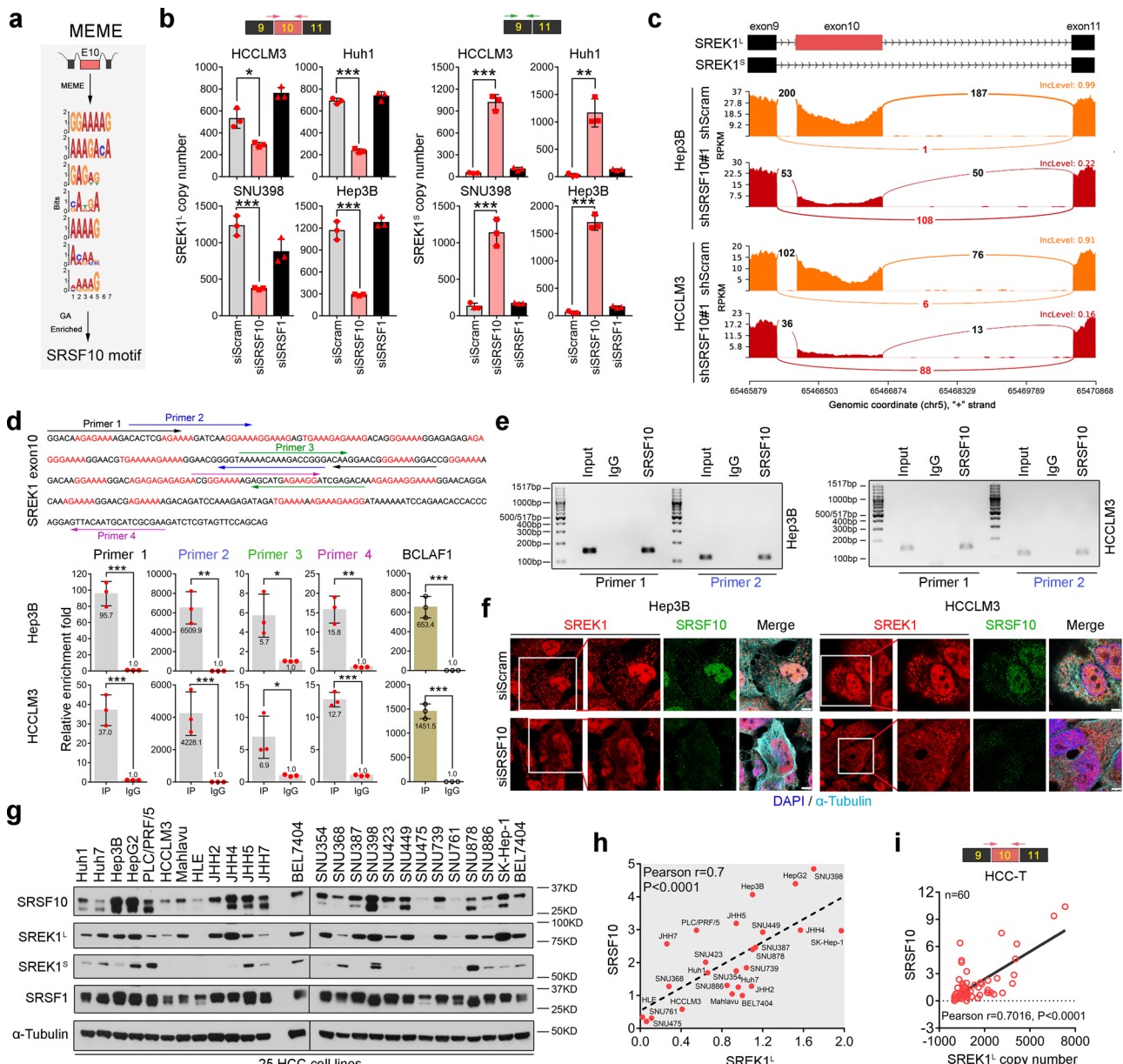

**Fig. 5 SREK1$^L$ is a direct AS target of SRSF10 in HCC cells. a** Identification of the SRSF10-binding motif in the exon 10-coding mRNA region by MEME motif analysis. **b** Expression analysis to determine the copy numbers of two SREK1 variants, SREK1$^L$ and SREK1$^S$, after SRSF10 or SRSF1 knockdown assays in four HCC cell lines ($n = 3$, mean ± SD). **c** Sashimi plots of exon 10 junction reads in stable SRSF10-knockdown or shScram-transfected cells. **d** Four sets of primers specific for the SREK1 exon 10 sequence (upper panel) were employed to detect exon 10 using real-time PCR (lower panel, $n = 3$, mean ± SD) and are labeled with different colors. **e** DNA gel detection of SREK1$^L$ binding in the RIP by SRSF10 or IgG in Hep3B or HCCLM3 cells. **f** Confocal microscopy of endogenous SREK1 (shown in red) and its colocalization with endogenous SRSF10 (stained with antibody against SRSF10, shown in green), microtubules (stained with antibody against α-tubulin, shown in purple) and nuclei (stained with DAPI, shown in blue) in siSRSF10- or siScram-transfected Hep3B and HCCLM3 cells, scale bar = 3 μm. **g** Immunoblot analysis of SRSF10, SRSF1, SREK1$^L$ and SREK1$^S$ in 25 HCC cell lines. Pearson correlation analysis of **h** the expression of SREK1$^L$ with SRSF10 protein expression in 25 HCC cell lines, and **i** the expression of SRSF10 with SREK1$^L$ in 60 HCC tumors (HCC-T) tissues. *$p < 0.05$; **$p < 0.01$; ***$p < 0.001$, two-tailed, unpaired $t$ test is used for (**b**, **d**). Source data are provided as a Source Data file.

to investigate whether SRSF10 is involved in CSCs. The data suggested that the knockdown of SRSF10, like that of SREK1$^L$, could markedly inhibit the renewal and growth of CSCs in HCC (Fig. 6e). To investigate the function of SRSF10 in vivo, we determined that the stable knockdown of tumor SRSF10 significantly delayed the tumorigenesis of both Hep3B and HCCLM3 cells in xenograft mouse models (Fig. 6f–h).

Recently, an SRSF10 inhibitor, 1C8, was developed and shown to inhibit HIV infection regulated by SRSF10 inhibition[20]. Here, 1C8

was employed, and its inhibition of SRSF10 in Hep3B cells was tested. 1C8 could significantly inhibit SREK1$^L$ expression in Hep3B cells (Fig. 6i) and suppress cell growth and colony formation in a dose-dependent manner in Hep3B cells (Fig. 6j, k). These data confirm that SRSF10 is a critical driver for hepatocarcinogenesis.

**RNA processing factors associated with SRSF10 also impact SREK1 expression in HCC cells.** The deregulation of splicing to drive carcinogenesis and/or metastasis frequently involves the

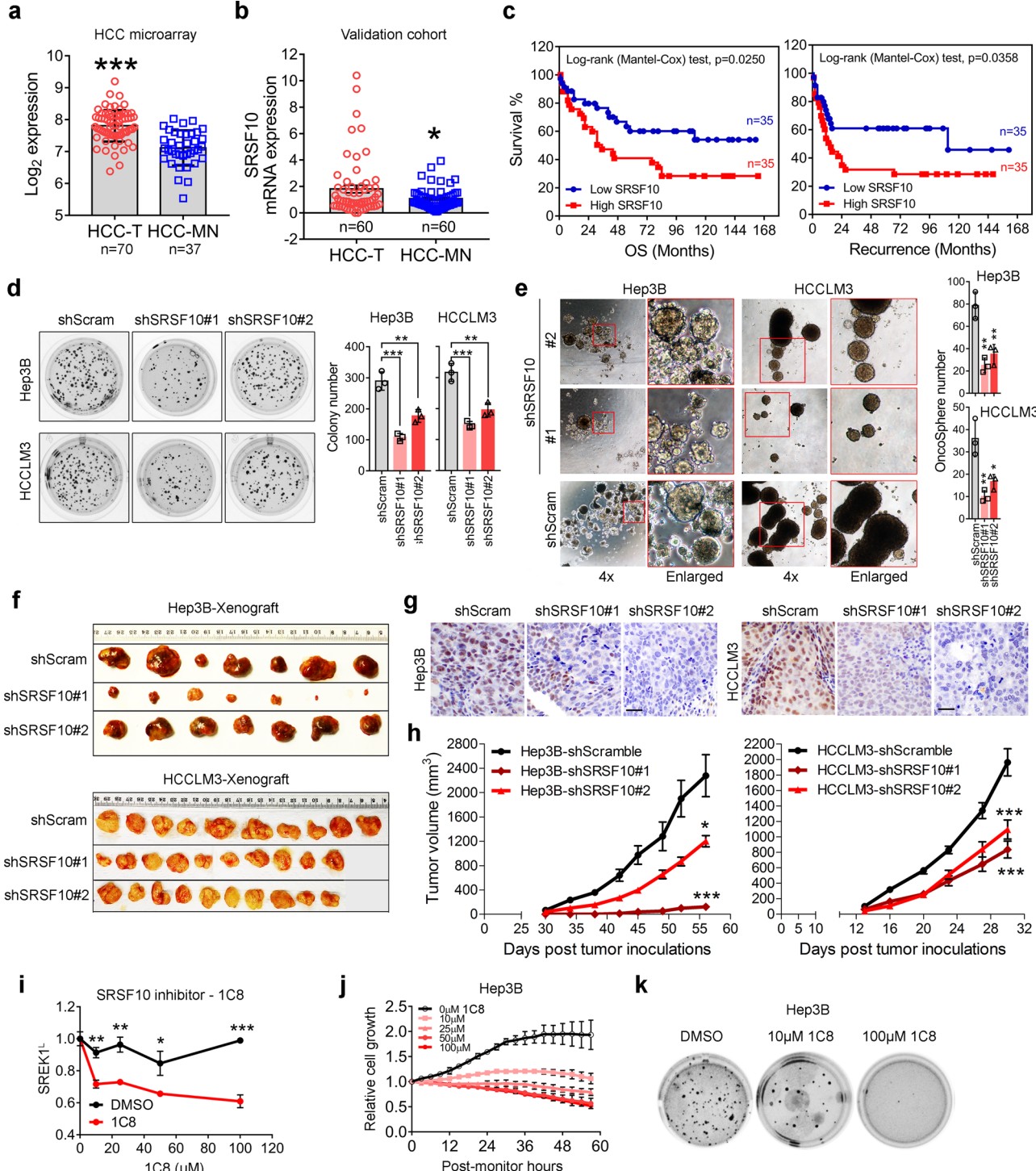

**Fig. 6 SRSF10 is associated with poor prognosis of HCC patients and acts as an oncogenic driver in HCC.** Expression of SRSF10 in HCC-T and HCC-MN using **a** a microarray database generated in our laboratory[35] or **b** PCR assay on a HCC cohort composed by 60 pairs HCC tissues (two-tailed, paired t test). **c** Comparison of OS and recurrence between HCC patients with high or low SRSF10 expression analyzed by our microarray database[25, 35]. **d** Anchorage-independent soft agar colony formation assay ($n = 3$) or **e** the oncosphere formation assays of the effect of SRSF10 or scramble control stable knockdown in HCCLM3 and Hep3B cells ($n = 3$). The representative tumor pictures (**f**), immunohistochemical staining (scale bar = 100 μm) (**g**), and tumor volume (**h**) of the mouse subcutaneous tumorigenic assay of the effect of SRSF10 or scramble control stable knockdown on the tumorigenesis of the Hep3B ($n = 7$) or HCCLM3 ($n = 10$) cells. Mean ± SEM; *$p < 0.05$; ***$p < 0.001$. **i** The inhibition of the 1C8, an SRSF10 inhibitor, on SREK1$^L$ expression in Hep3B cells. **j** The cell growth and **k** the anchorage-independent soft agar colony formation assays of 1C8 on the inhibition of Hep3B cells ($n = 3$). Mean ± SD, *$p < 0.05$; **$p < 0.01$; ***$p < 0.001$, two-tailed, unpaired t test is used for (**a**, **d**, **e**, **h**, **i**) for; *$p < 0.05$; **$p < 0.01$; ***$p < 0.001$. Source data are provided as a Source Data file.

reconstruction of aberrant splicing regulatory complexes[9,36]. To investigate the potential intrinsic regulatory mechanism(s) of SRSF10 in SREK1 splicing in HCC cells, the stable isotope labeling with amino acids in cell culture (SILAC) proteomic strategy was employed to reveal endogenous factors that interact with SRSF10 by IP using two commercial SRSF10 antibodies and mass spectrometry (Fig. 7a). We identified 476 common potential endogenous interactors of SRSF10, and 30 of them were further shortlisted by setting a cut-off of an H/L ratio >1.7 (Fig. 7b). IPA cellular pathway analysis showed that 26.67% of the potential interactors were related to mRNA splicing, while 16.67% and 16.67% were related to metabolism and transcription, respectively (Fig. 7c), indicating that the function of SRSF10 in Hep3B cells is mainly in splicing and/or transcriptional regulation. Eight interactors related to AS and three interactors related to transcription and/or metabolism were selected for further investigation (Fig. 7c). To investigate whether SREK1 splicing was regulated by these potential interactors, SREK1 splicing was evaluated by quantification of the ratio of SREK1$^L$ and SREK1$^S$ expression by western blotting (Fig. 7d) and real-time PCR (Fig. 7e). As shown by these data, six out of the eleven interactors studied were found to be involved in SREK1 splicing potentially and they were: ELAVL1, MAGOH, PABPC1, SF3B6, MAGOHB and SFPQ (Fig. 7f).

To decipher the molecular roles of these six interactors, an independent SRSF10 IP assay was performed in both Hep3B and HCCLM3 cells (Fig. 7g). The results suggested that only ELAVL1, PABPC and MAGOH/MAGOHB (two proteins with similar sequences), but not PECR, SFRQ and SF3B6, form a complex or multiple independent complexes with SRSF10 to regulate SREK1 splicing (Fig. 7g). To further confirm that these three interactors bind to SREK1 pre-mRNA, RIP analysis was performed, which showed that all the antibody targeting three interactors could recover SREK1$^L$ pre-mRNA (Fig. 7h), indicating that SRSF10 can interact with ELAVL1, PABPC and/or MAGOH/MAGOHB to regulate SREK1 splicing in HCC cells.

**The SRSF10-SREK1 splicing signaling regulates HCC tumorigenesis.** We demonstrated that SREK1 may act as an oncogenic downstream effector of SRSF10 by forming a SRSF10/SREK1 splicing signaling to promote hepatocarcinogenesis. To confirm it, SREK1$^L$ or mCherry (as a control) was re-expressed in SRSF10-knockdown or Scram-transfected HCCLM3 cells (Supplementary Fig. 7a) and assessed by oncogenic colony formation assay (Fig. 8a and Supplementary Fig. 7b). Stable knockdown of SRSF10 significantly attenuated the colony number, but when SREK1$^L$ was re-expressed in the cells, the knockdown of SRSF10 did not significantly inhibit colony formation (Fig. 8a and Supplementary Fig. 7b). This indicated that SREK1$^L$ is a direct functional effector of SRSF10 in HCC. We also showed that, while SRSF10 knockdown significantly attenuated the tumorigenesis of HCCLM3 xenografts, re-expression of SREK1$^L$ reversed this inhibitory effect (Fig. 8b, c and Supplementary Fig. 7c). When the expression of SRSF10, SREK1, Ki-67 and active Caspase 3 in the xenografts was analyzed by IHC staining, SRSF10 expression was reduced in the SRSF10-knockdown xenografts. SREK1 was mostly localized in the nuclei of Scram-transfected xenograft cells, and SRSF10 knockdown significantly reduced the expression of nuclear SREK1 and Ki-67 but increased the expression of cytoplasmic SREK1 and active Caspase 3 (Fig. 8d), confirming that SRSF10 knockdown inhibited SREK1 nuclear localization and cell proliferation and promoted apoptosis in xenograft tumors. When SREK1$^L$ was re-expressed in HCCLM3 cells, SRSF10 knockdown failed to regulate the expression of nuclear SREK1, Ki-67 and active Caspase 3 (Fig. 8d), indicating that re-expression of SREK1$^L$ could rescue the inhibitory effect of SRSF10 knockdown on tumorigenesis.

To verify the expression changes of the downstream factors in SRSF10-SREK1$^L$ signaling in xenograft, the gene expression was evaluated by PCR in tumors (Fig. 8e). Stable knockdown of SRSF10 in xenograft could significantly inhibit the expression of SRSF10, SREK1$^L$, B-T and TXNDC5, while promoting the expression of miR-30c-5p and miR-30e-5p (Fig. 8e). Re-expression of SREK1$^L$ in xenograft could markedly promote the expression of SRSF10, B-T and TXNDC5, but significantly inhibit the expression of miR-30c-5p and miR-30e-5p (Fig. 8e).

Our data confirm that the splicing factor SRSF10 could sustain the high expression level of SREK1$^L$ to inhibit the NMD of B-T, which in turn acts as a ceRNA to promote the oncogenic drivers —SRSF10 and TXNDC5 expression by inhibiting miR-30c-5p and miR-30e-5p (Fig. 8f). Thus, an SRSF10/SREK1L/B-T oncogenic signaling loop has been revealed here for hepatocarcinogenesis (Fig. 8f).

## Discussion

Aberrant AS regulators have recently been implicated in hepatocarcinogenesis[37–44]. Therefore, it may be fruitful to decipher key oncogenic splicing network to enable the development of specific inhibitors targeting splicing. We have demonstrated that the oncogenic SRSF10/SREK1 splicing signaling is crucial for carcinogenesis through its regulation of NMD of B-T in HCC cells. SRSF10 was recently reported to be a druggable splicing regulator, and its reported inhibitor, 1C8, has been shown to inhibit HIV replication[20]. In this study, we demonstrated that 1C8 was also effective in inhibiting the growth of HCC cells, suggesting the involvement of SRSF10 phosphorylation in regulating SRSF10 activity in HCC. SRSF10 has been reported to have tissue- or cell-type-specific roles during physiological processes and disease development[19,21–23]. One of the possible mechanisms for these roles is through the regulation of diverse spectra of SRSF10 interactors under different cell types and physiological processes. Currently, little is known about how SRSF10 modulates its molecular organization to generate effective molecular functional complexes. Here, we have provided evidence of an HCC-related functional complex of SRSF10 with ELAVL1, PABPC and MAGOH/MAGOHB that modulates SREK1 splicing.

Recently update on SRSF10 and TXNDC5 have indicated their diverse and signaling regulatory roles in cancers. SRSF10 can act as a sequence-dependent splicing factor to regulate the AS of BCLAF1 and mIl1RAP to promote the growth of cancer cells in colorectal cancer and cervical cancer[22,23]. In Head and Neck Cancer, SRSF10 regulates the splice variants of BCL2 Like 1 and Pyruvate kinase M to promote tumorigenesis[45]. In addition, SRSF10 may also involve in modulating circular RNA biogenesis to regulate glioma angiogenesis[46]. TXNDC5 has been reported as a critical mediator of fibrosis by enhancing TGF-β1 signaling in cardiac, renal and lung[47–49]. In prostate cancer, TXNDC5 can directly interact with and stabilize the AR protein to promote Castration-resistant prostate cancer development and growth[34]. TXNDC5 is also involved in PI3K/AKT signaling pathway to promote ESCC cell proliferation and invasion[50].

The NMD pathway has been indicated in cancer, but its roles and targets in HCC are largely unknown[51–53]. Tumors can manipulate NMD to downregulate gene expression by promoting specific mutations that cause the destruction of key tumor-suppressor mRNAs, and occasionally, tumors can also adjust NMD activity to enable their adaptability to different microenvironments[53]. We revealed that a noncoding candidate NMD gene, B-T, is upregulated and oncogenic in HCC. B-T was found to act as ceRNA to inhibit the endogenous miR-30c-5p and miR-30e-5p. Interestingly, the B-T/microRNA regulatory network would further in turn control the gene expression of many splicing factors, such as SRSF10. This indicates that the regulation of NMD pathway might be crucial for the

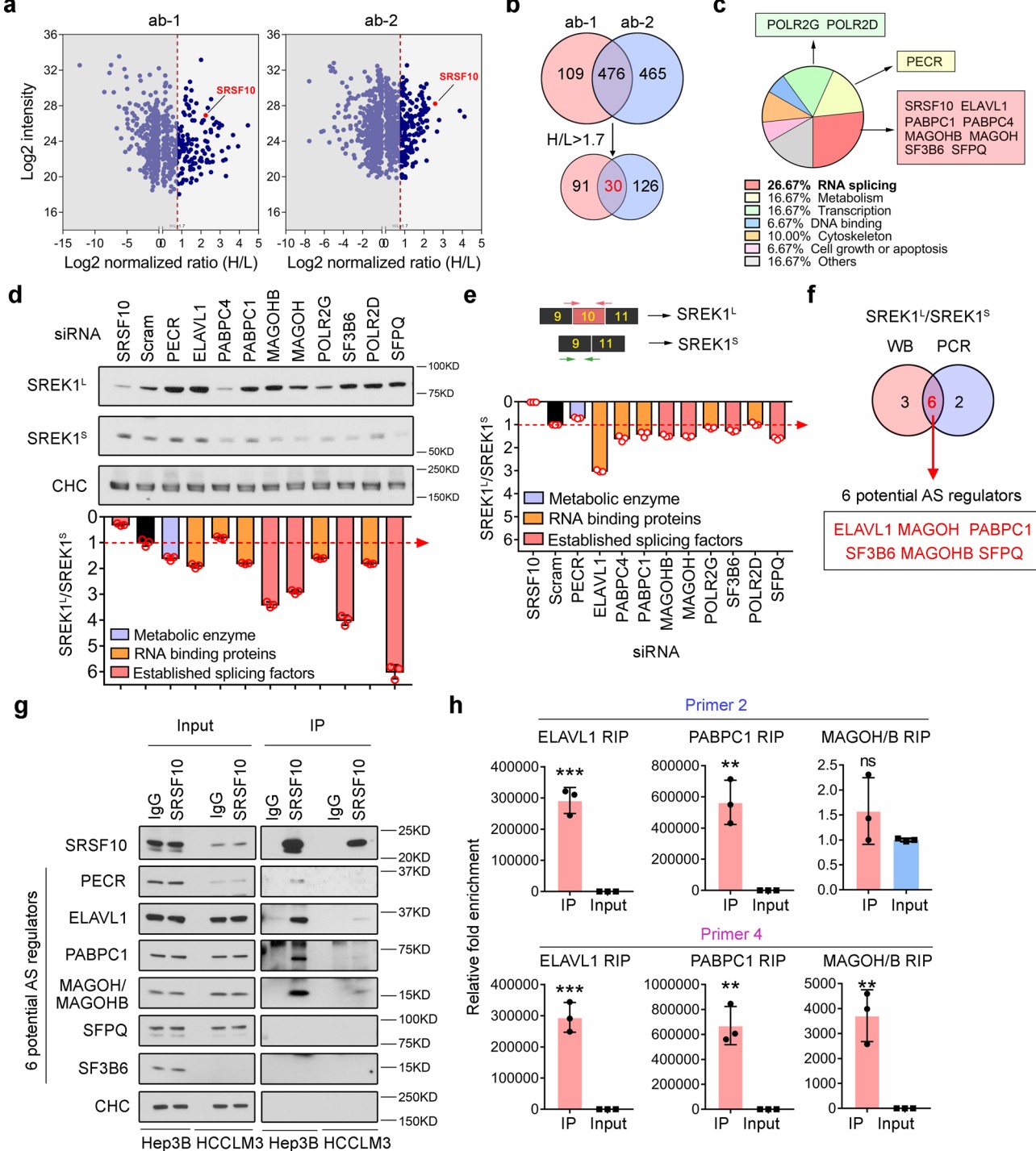

**Fig. 7 Dissection of the SRSF10-associated proteins involved in regulating SREK1$^L$ in HCC cells. a** Dot plots of endogenous SRSF10 interactors identified by the SILAC approach using two commercial antibodies against SRSF10 in Hep3B cells. The SRSF10 bait protein was indicated by red dots, and the dashed line indicates the cut-off of H/L ratio. **b** Venn diagram analysis identified 30 SRSF10 potential interactors identified by both antibodies with the cut-off of H/L > 1.7. **c** Functional annotation of the 30 interactors indicating that 26.67% are related to RNA splicing, and the interactors identified in the top 3 pathways are labeled in colored boxes. **d** Immunoblot analysis of SREK1$^L$, SREK1$^S$ and Clathrin heavy chain (CHC) following the knockdown of each of the identified SRSF10 interactors in Hep3B cells. The ratios of SREK1$^L$/SREK1$^S$ were quantified, and the ratio in cells with a scramble control was set as the cut-off indicated with a dotted red line ($n = 3$). **e** Ratio of SREK1$^L$/SREK1$^S$ mRNA expression post the knockdown of each SRSF10 interactors, and expression in cells with a scramble control was set as the cut-off indicated with a dotted red line ($n = 3$). **f** Venn diagram analysis identified six common SRSF10 splicing-related interactors determined by both immunoblotting and real-time PCR. **g** Independent endogenous immunoprecipitation validation of the six SRSF10 splicing-related interactors in Hep3B and HCCLM3 cells. **h** RIP analysis demonstrated that three of the validated SRSF10 interactors binding with SREK1$^L$ mRNA in Hep3B cells ($n = 3$). Mean ± SD, two-tailed, unpaired $t$ test is used for (**d**, **e**, **h**), **$p < 0.01$; ***$p < 0.001$. Source data are provided as a Source Data file.

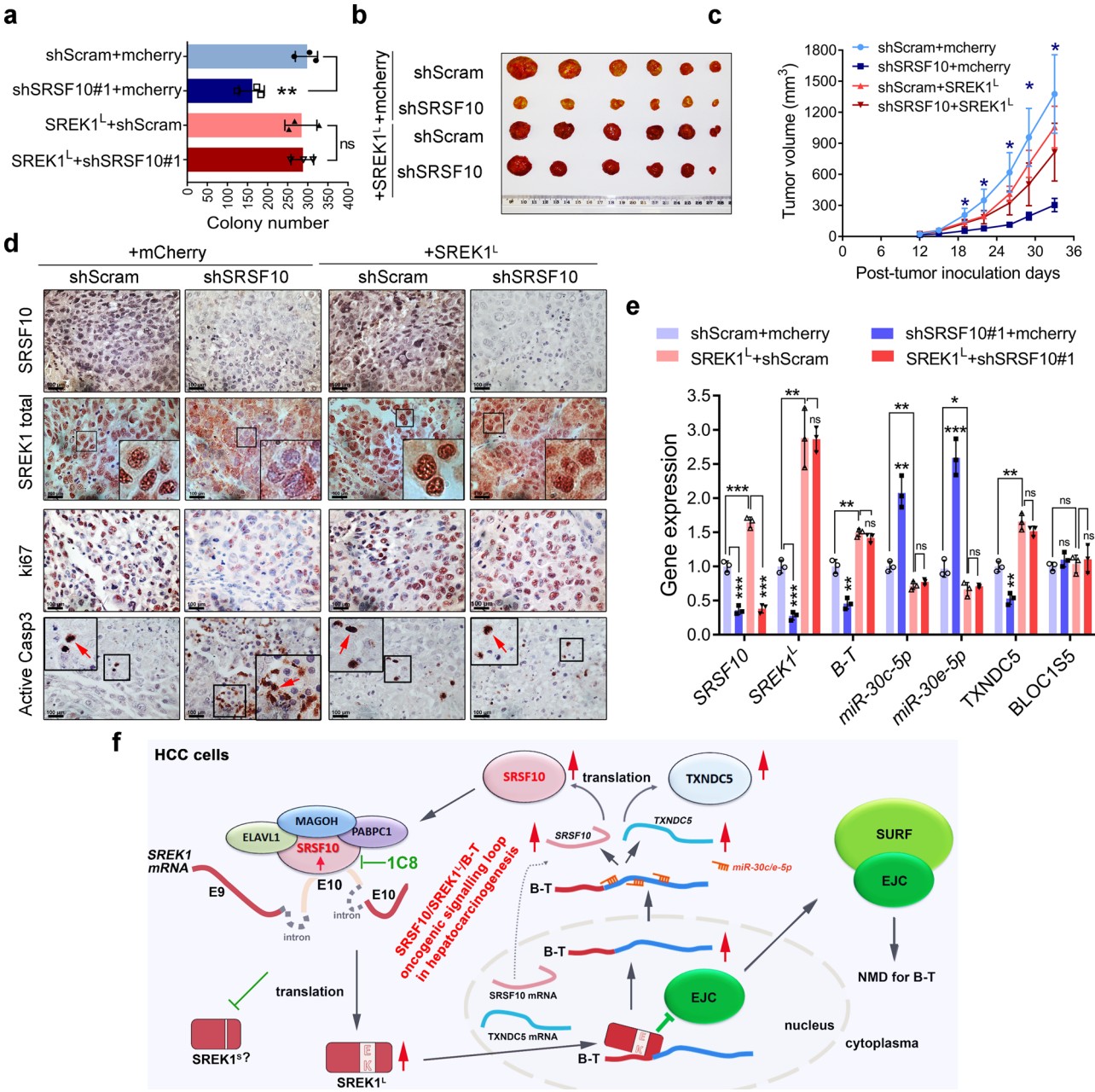

**Fig. 8 SREK1$^L$ is an oncogenic downstream effector of SRSF10 to promote the carcinogenesis of HCC cells, and the oncogenic SRSF10/SREK1$^L$/B-T signaling loop is crucial for hepatocarcinogenesis. a** The quantification of anchorage-independent soft agar colony formation assays of stable knockdown of SRSF10 or the combination of the forced expression of SREK1$^L$ in HCCLM3 cells ($n = 3$, mean ± SD, two-tailed, unpaired $t$ test is used, **$p < 0.01$; ns non-significant). **b** The represented xenograft tumors and **c** mouse xenograft tumorigenic assays in stable SRSF10-knockdown, scramble-transfected HCCLM3 cells or stable SRSF10-knockdown HCCLM3 cells following SREK1$^L$ re-expression ($n = 6$, mean ± SEM, two-tailed, unpaired $t$ test is used, *$p < 0.05$). **d** Immunohistochemical staining of HCCLM3 xenografts for SRSF10, SREK1, Ki-67 and active Caspase 3 (scale bar = 100 μm). **e** A panel of genes expression in xenograft tumors evaluated by real-time PCR ($n = 3$, mean ± SD, two-tailed, unpaired $t$ test is used, *$p < 0.05$; **$p < 0.01$; ***$p < 0.001$; ns non-significant). **f** A proposed regulatory model showing how the oncogenic SRSF10/SREK1$^L$/B-T signaling loop in HCC cells promoting hepatocarcinogenesis. Source data are provided as a Source Data file.

maintenance of activity of AS signals in tumors, especially for those with key driver roles in hepatocarcinogenesis.

The deregulation of AS regulators in cancer is frequently associated with abnormal expression and/or activities via modifications or aberrant interactions that maintain the potential for cancer cell growth. Thus, a group of deregulated splicing regulators, as exemplified by SRSF10, could be useful potential therapeutic targets for HCC.

## Methods

**Tissues, antibodies and microarrays.** The collection of tissues from HCC patients was approved by the SingHealth Centralized Institutional Review Board, and all tissues studied were provided by the SingHealth Tissue Repository. Written informed consent was obtained from all the participating patients, and relevant clinical and histopathological data provided to the researchers were anonymized.

The antibodies used for this study are listed in Supplementary Table 5. Total RNA from 70 HCC tumors and 37 nontumor tissues were extracted using TRIzol® Reagent (Life Technologies, Carlsbad, CA) according to the manufacturer's protocol. Five micrograms of purified total RNA were reverse transcribed, labeled

with biotin, hybridized onto the GeneChip Human Genome U133 Plus 2.0 Array (Affymetrix, Santa Clara, CA) to determine the expression level of each gene as previously described[25,35,54,55]. Raw expression data were normalized using Gene Chip Robust Multiarray Averaging method (Partek Genomics Suite® version 7.0, Partek Incorporated Inc, MO, USA). The microarray data have been deposited in the ArrayExpress public database with accession numbers E-MEXP-84 and E-TABM-292.

**SiRNAs, constructions, transfection and cell lines.** The following siRNAs targeting different human genes were constructed: SMARTpool ON-TARGETplus Non-targeting pool siRNA against *scramble* (D-001810-10-50), siRNA against *SRSF10* (L-012914-02-0005), siRNA against BLOC1S5-TXNDC5 (R-188208-00-0005) from GE health Dharmacon, and siRNA against *SREK1* siE10#1 (Sequence 5'-3': GGAUCGAGACAAAGAGAAG), siE10#2 (Sequence 5'-3': GACA-GAUCCAAAGAGAUAG), siE9E11#1 (Sequence 5'-3': UACGCCUUUCCCGU-GAAUGCG), siE9E11#2 (Sequence 5'-3': CUUUCCCGUGAATGCGAACGACT), siE9E11#3 (Sequence 5'-3': UCCUACGCCUUUCCCGTGAATGC) and siRNA against *UPF1* siUPF1 (Sequence 5'-3': GAUGCAGUUCCGCUCCAUU). Sequences will be made available on request for the commercial siRNAs. The Lipofectamine RNAiMax (Life Technologies, 13778-150) was used for siRNA transfection. The final concentrations of RNA duplex and miRNA inhibitor were 50 and 200 nM, respectively. Transfection with plasmid DNA alone or together with RNA duplex was performed using Lipofectamine 2000 (Invitrogen, 11668019).

Flag-SREK1[L] (EXT3538Lv102), TXNDC5 (EX-Z9676-Lv244) and pBLOC1S5-TXNDC5 (EX-Y1877-Lv156) were purchased from Genecopoeia. The SREK1[L]-4KA mutations were generated by a mutagenesis kit (Agilent, 200518) from Agilent Technologies. Transfection of cells with miRNA mimics, inhibitors and control mimics/inhibitors and/or plasmids was performed with Lipofectamine 3000 (Invitrogen, L3000015) according to the manufacturer's protocol. The *mirVana*™ miRNA inhibitors or mimics for miR-30c-5p (inhibitor Assay ID: MH11060; mimics Assay ID: MC11060), miR-30e-5p (inhibitor Assay ID: MH10037; mimics Assay ID: MC10037) and NC (inhibitor 4464076; mimics 4464058) were purchased from Thermo Fisher Scientific Inc., MA, USA.

The Hep3B (HB-8064), HepG2 (HB-8065), PLC/PRF/5 (CRL-8024), SNU449 (CRL-2234), SK-Hep-1 (HTB-52) and HEK-293T (CRL-3216) were obtained from American Tissue Culture (ATCC). Huh7 (JCRB0403), Huh1 (JCRB0199), HLE (JCRB0404), JHH2 (JCRB1028), JHH4 (JCRB0435), JHH5 (JCRB1029), JHH7 (JCRB1031) were obtained from Japanese Collection of Research Biosources Cell Bank (JCRB). SNU354 (KCLB00354), SNU368 (KCLB00368), SNU387 (KCLB00387), SNU398 (KCLB00398), SNU423 (KCLB00423), SNU449 (KCLB00449), SNU739 (KCLB00739), SNU761 (KCLB00761), SNU878 (KCLB00878), SNU886 (KCLB00886) cell lines were obtained from Korean Cell Line Bank (KCLB). The HCCLM3, Mahlavu and BEL7404 cells were gifts from Dr John M Luk. All cell lines were cultured in DMEM supplemented with 10% fetal bovine serum. Mycoplasma test was done by Shik Nie Kong at NCCS.

**Construction of stable cells.** The MISSION sh lentivirus system was used to generate stable knockdown HCC cells in our experiments. Lentiviral cloning vectors pLKO.1-hPGK-Puro-CMV-tGFP was used to deliver various shRNA constructs: shE10#1 (Sequence 5'-3': GGAUCGAGACAAAGAGAAG), shE10#2 (Sequence 5'-3': GACAGAUCCAAAGAGAUAG), shSRSF10#1 (Sequence 5'-3': GCUGAAGACGCUUUACAUAAUU), shSRSF10#2 (Sequence 5'-3': GCGU-GAAUUUGGUCGUUAU), shScram (Sequence 5'-3': UGGUUUACAUGUUGU-GUGA). Sh*Scram* was purchased from Sigma-Aldrich (Sigma-Aldrich, SHC016). Lentivirus was prepared according to the supplier's instructions and employed to infect cells. The cells were sub-cultured to 10% confluence in a medium containing puromycin (Sigma-Aldrich, P8833) 1.5 μg/ml for Hep3B cell and 2.5 μg/ml for HCCLM3 cell. Antibiotic-resistant clones were picked and passaged in a medium containing puromycin. The level of the specific knockdown protein was assessed by western blot assays and/or real-time PCR.

**Tumorigenic study.** All protocols involving the use and handling of laboratory animals were approved by the SingHealth Institutional Animal Care and Use Committee (IACUC). Five- to seven-week-old male BALB/c-Nude or C57BL/6 mice were purchased from INVIVOS PTE Ltd and housed in the animal facility at NCCS with the conditions: temperature range 19–26 °C, temperature fluctuations within bandwidth 3–4 °C, humidity range 50–70%, light intensity 300–350 lux at 1 m above the floor, uniform 12 h light/dark daily lighting cycle, ventilation 15–20 air change per hour. Stably transfected HCCLM3 or Hep3B cells were resuspended in PBS and subcutaneously implanted into the left and right flanks ($5 \times 10^6$ cells per flank) of mice. Tumor volume was calculated according to the formula volume = $0.5 \times$ length $\times$ width$^2$ as described previously[56,57]. The mice should be euthanized once the xenograft volume reaches 2000 mm$^3$ that has been permitted by IACUC with the Ref No.: 2016/SHS/1198.

**Soft agar colony formation assay.** Stable knockdown cells were counted and seeded in six-well plates (for Hep3B and HCCLM3, 10,000 and 20,000 cells were seeded accordingly) in a growth medium containing 0.7% agar (2 ml per well) on top of a layer of growth medium containing 1.4% agar (1 ml per well). A growth

medium (1 ml) with 10% FBS (Fetal bovine serum) was added on top of the agar. The cell suspension was plated and cultured in a 37 °C incubator for 2–3 weeks. After that, the colonies were stained with MTT (Sigma-Aldrich, M2003), counted and monitored.

**Oncosphere formation assay.** For oncosphere formation assay, 5000 HCCLM3 or 5000 Hep3B stable transfected cells were seeded on ultra-low attachment culture dishes (Corning, 3261) in a serum-free medium. DMEM/F12 serum-free medium (Invitrogen, 11320033) contained 2 mM L-glutamine, 1%sodium pyruvate (Invitrogen, 11360070), 100 μg/ml penicillin G, and 100 U/ml streptomycin supplemented with 20 ng/ml epithelial growth factor (Invitrogen, RP-10914), 10 ng/ml fibroblast growth factor-2 (Invitrogen, PHG0261), N2 (Thermo Fisher Scientific, 17502001), and B27 (Thermo Fisher Scientific, 17504001). Cells were incubated in a CO$_2$ incubator for 1–2 weeks, and the number of oncosphere cells (diameter >100 μm) were counted using a stereomicroscope (NIKON, Tokyo, Japan).

**BrdU proliferation and wound-healing migration assays.** HCC cell lines were treated with siRNAs targeting *SRSF10* or *scramble* control genes. After 10 h, the cells were digested with trypsin and 10,000–20,000 cells were seeded in 96-well plates. The BrdU proliferation assay kit (Cell Signalling Technology, 6813) was used.

Wound-healing assays were done as previously described[25,54,55]. Briefly, stable shSRSF10-, shSREK1- or scramble control-knockdown cells were grown to confluent and a 100 μl yellow tip was used to make a scratch in the middle of the well, gently wash once by PBS and 1% FBS serum reduced DMEM was added to limit cell proliferations throughout the assay. After incubation, photos of different wound regions were taken and the size of the wound was calculated by Image-J software version 1.51j8 (National Institutes of Health, USA).

**Nuclear and cytoplasmic protein extraction.** For detection of the nuclear and cytoplasmic SREK1, NE-PER@ Nuclear and Cytoplasmic Extraction kit (Thermo Scientific Pierce, 78833) was used for HCCLM3 cells by siSRSF10 and siScramble knockdown treatment as described before[25]. Briefly, fresh cells ($5 \times 10^6$ cells) in 10 cm dish were collected and processed for cell fractionation analysis. Briefly, wash the cell with 2 ml ice-cold PBS by pipetting and repeat two more times, Resuspend the cell pellet in 1 ml ice-cold Extraction Lysis Buffer by pipetting. Incubate for 10 min (min) at 4 °C on an end-over-end shaker. Centrifuge the lysate at 1000 × g for 10 min at 4 °C. Carefully transfer the supernatant into a fresh microcentrifuge tube. Store on ice. This fraction primarily contains cytosolic proteins. Resuspend the pellet in 1 ml ice-cold RIPA Buffer by pipetting. Incubate for 30 min at 4 °C. Centrifuge the suspension at 12,000 × g for 10 min at 4 °C. Carefully transfer the supernatant nuclear protein into a fresh microcentrifuge tube and Store on ice for further protein quantification analysis.

**RNA immunoprecipitation (RIP).** RIP was done using the Millipore MagnaRIP Kit (Millipore, 17-700) and according to the protocol provided. Antibodies used were anti-SRSF10 (Sigma-Aldrich, HPA053831), anti-HuR (Santa Cruz, sc-5261), anti-MAGOH (Santa Cruz, sc-271405), anti-PAPBC1 (Santa Cruz, sc-32318), anti-SRSF1 (Santa Cruz, sc-33652). In brief, for each IP reaction, Hep3B cells were grown in 15-cm dish to 80% confluence. Cells were washed with ice-cold PBS, followed by scrapping and pelleted down at 1500 RPM (4 °C) for 5 min. The pellet was resuspended in the Lysis Buffer (115 μl) provided with the kit and incubated on ice for 5 min. The lysate was frozen and stored at –80 °C to complete the lysis. Before IP reaction, the cells were thawed and centrifuged at 14,000 × g for 10 min at 4 °C. For each IP reaction, 5 μg Antibody and 50 μl magnetic beads were coupled at room temperature for 30 min. A total of 100 μl of thawed lysate was used for each IP reaction, 10 μl of lysate was removed for input and stored in –80 °C till RNA elution. IP was done overnight at 4 °C with rotation and then washed four times with RIP Wash Buffer. Washed beads and Input lysate were resuspended in 150 μl RIP Wash Buffer, supplemented with 0.1% SDS and 180 μg Proteinase K incubated for 30 min at 55 °C. After proteinase digestion, RNA was extracted by phenol-chloroform-isoamylalcohol and ethanol precipitation. SensiFAST™ cDNA Synthesis Kit (Meridian Bioscience, BIO-65054) was used to convert RNA to cDNA. Real-time PCR SensiFAST™ SYBR Kit (Meridian Bioscience, BIO-98020) was used to detect *SREK1* transcript using the following primers: SREK1 RIP-1, F: 5'-GACAAGAGAAAAGACACTCGAGAA-3', R: 5'-GTCCTTTTCCCGT TCCTTG-3'; SREK1 RIP-2, F: 5'-CACTCGAGAAAAGATCAAGGAAA-3', R: 5'-CC CGGTCTTTGTTTTTACCC-3'; SREK1 RIP-3, F: 5'-AAAAACAAAGACCGGGACA A-3', R: 5'-TGTCTCGATCCTTCTCATGCT-3' and SREK1 RIP-4, F: 5'-GACAAGA GAAAAGACACTCGAGAA-3', R: 5'-GTCCTTTTCCCGTTCCTTG-3'.

**SILAC study of SRSF10 interactome.** Hep3B cells were stably transfected with shSRSF10 or shScramble followed by puromycin selection before SILAC. Stably transfected Hep3B cells were cultured in DMEM for SILAC (Thermo Fisher Scientific, A33822) supplemented with 10% dialyzed FBS and 1% Penicillin/Streptomycin containing either normal isotopes of L-lysine-(12C614N2) (K0) and L-arginine-(12C614N4) (R0) (K0R0-"light" medium) for shSRSF10 or stable isotope L-lysine-(13C615N2) (K8) and L-arginine-(13C615N4) (R10) (R10K8-"heavy" medium) in the case of shScramble cells. The whole process of SILAC was performed as described[54]. Briefly, cells from 8 × 15 cm culture dishes were lysed in IP

lysate buffer (Pierce, 87788) with the addition of proteinase inhibitor (Sigma-Aldrich,, P8340), phosphatase inhibitor (Sigma-Aldrich, P0044), Pierce Universal Nuclease (Pierce, 88701) and DTT (Sigma-Aldrich, 43815). A total of 80 μl magnetic beads from Dynabeads™ Protein G Immunoprecipitation Kit (Thermo Fisher Scientific, 10007D) were washed with IP lysate buffer and coupled with 40 μg SRSF10 antibody (Sigma-Aldrich, HPA053805) by incubating overnight at 4 °C. Thirty mg of total lysates from heavy or light medium-labeled cells were pre-cleaned by 20 μl magnetic beads each for 1 h, followed by IP with 40 μl antibody-coated beads for 4 h at 4 °C. Following IP, the samples were washed four times with IP Lysate buffer and eluted in 30 μl 1.5× NuPAGE LDS sample buffer with 0.1% DTT. Eluted samples were heated at 95 °C for 10 min. The IP eluate was separated by one-dimensional NuPAGE™ 4–15% Bis-Tris Protein Gel and stained with Colloidal Blue (Invitrogen, LC6025). Protein bands were cut and digested with trypsin. The samples were then analyzed with an Orbitrap Classic mass spectrometry (Thermo Fisher Scientific Inc., MA, USA). Identification and quantification of the sample products were performed using MaxQuant version 1.5.0.30.

**Immunoblotting (IB) and immunoprecipitation (IP).** Cell lysate was prepared on ice for 30 min in RIPA or IP lysis buffer (Thermo Fisher Scientific, 87788) with the addition of the protease inhibitor cocktail, phosphatase inhibitor cocktail, and DTT. The lysate was spun down and the protein concentration of the supernatant was determined using the BCA Protein assay kit (Thermo Fisher Scientific, 23225). Normally for IB, 20 μg total proteins with 1× LDS sample buffer were loaded for gel running. NuPAGE Novex 4–12% Bis-Tris protein gels (Life Technologies, NP0336PK2) were used for the IB.

Protein G Dynabeads were pre-cleaned and incubated with antibodies (listed in Supplementary Table 5) overnight in 5% BSA IP buffer at 4 °C. After which, the beads were washed with IP buffer (20 mM Tris pH 8, 10% glycerol, 150 mM NaCl, 0.1% NP-40, 0.1 mM EDTA) three times on a magnet where the beads migrated to the side of the tube facing the magnet. HCCLM3 cells, after different treatments as specified, were grown in a 15-cm culture dish to a density of 70–90% confluence and were lysed on ice with IP buffer for 30 min. Protein concentration was measured and 500–1000 mg of the protein lysate was added to the beads and incubated for 2 h at 4 °C. After which, the beads were washed six times. Finally, the proteins on the beads were boiled with 1× SDS loading buffer on a 95 °C heat block for 10 min. The samples were analyzed by western blotting and detected by specific anti-mouse (Abcam, ab99697) or anti-rabbit (Abcam, ab99617) secondary antibodies that recognize the light chain.

**RNA isolation, real-time PCR and EJC deposition assay.** The Hep3B, LM3 cells were collected by trypsinization, followed by RNA extraction using the RNEasy Mini kit (QIAGEN 74106). The concentrations of RNA were determined by Nanodrop One (Thermo Fisher Scientific Inc., MA, USA). The cDNAs were prepared by SensiFAST cDNA Synthesis Kit (Bioline, BIO-65054) and processed for real-time PCR detection with SensiFAST SYBR No-ROXd Kit (Bioline, BIO-98020) by CFX Connect Real-Time System (Bio-Rad Laboratories Inc., California, USA) using the primer sequences listed in Supplementary Table 6. Reverse transcription of 2 ng/μl of RNA into cDNA was carried out using TaqMan miRNA Reverse Transcription Kit (Thermo Fisher Scientific, 4366597) specific for each miRNA of interest (assay ID: 000419 for hsa-miR-30c-5p, and 002223 for hsa-miR-30e-5p) and the reference gene small nuclear RNA U6 (U6snRNA) (assay ID: 001973) were ordered from Thermo Fisher Scientific Inc., MA, USA.

Nuclear extracts were prepared from Hep3B cells as described[58]. Briefly, the Hep3B nuclear pellet obtained from the low-speed centrifugation of the homogenate was subjected to second centrifugation for 20 min at 25,000 g, to remove residual cytoplasmic material and this pellet was resuspended in 3 ml of buffer C [20 mM HEPES (pH 7.9), 25% (v/v) glycerol, 0.4 2 M NaCl, 1.5 mM MgCl, 0.2 mM EDTA, 0.5 mM phenylmethylsulfonyl fluoride (PMSF) and 0.5 mM DTT] per 10^9 cells with a Kontes all glass Dounce homogenizer (10 strokes with a type B pestle). The resulting suspension was stirred gently with a magnetic stirring bar for 30 min and then centrifuged for 30 min at 25,000 g and the clear supernatant was dialyzed against 50 volumes of buffer D [20 mM HEPES (pH 7.9), 20% (v/v) glycerol, 0.1 M KCl, 0.2 mM EDTA, 0.5 mM PMSF, and 0.5 mM DTT] for 5 h. The dialysate was centrifuged at 25,000 g for 20 min and the resulting precipitate discarded. The supernatant, designated the nuclear extract, was frozen as aliquots in liquid nitrogen and stored at −80° for further use. Biotinylated B-T or B-T ΔBS pre-mRNAs were prepared according to the Riboprobe® Combination Systems (Promega, P1460). Pre-mRNAs were gel purified, phenol-chloroform extracted, ethanol precipitated, and resuspended in nuclease-free water for further use.The SREK1^L binding sites on the B-T exons junction region of the mRNA was predicted by RNA-Protein Interaction Prediction and only the binding sequence "GCTAATGACTCAGTCTGTAG" (SVM reading >0.5, 32~51 bp upstream of the exons junction site of *BLOC1S5* and *TXNDC5*) was determined and selected for the test. In vitro EJC deposition reactions with biotinylated mRNA were performed as described before[59]. Briefly, the reaction consisted of 13.6 μl of 10× SP buffer [5 mM adenosine 5′-triphosphate, 200 mM creatine phosphate, 24 mM MgCl2], 5 nM biotinylated mRNA, 0.7 U/μl SUPERaseIN (Thermo Fisher Scientific, AM2694), 60 μl of Hep3B nuclear extract, and 25 μl each of two HEK-293T whole-cell extracts expressing Flag-SREK1^L or control vector (containing 3× FLAG on the N terminus). Reaction mixtures were then diluted with 564 μl of HNT buffer [20 mM

Hepes-KOH, 150 mM NaCl, 0.5% (wt/vol) Triton X-100] and incubated with M-280 streptavidin-coated magnetic beads (Thermo Fisher Scientific, 112.06D) at 37 °C for 1 h. Beads were washed ten times with HNT buffer at 37 °C; RNA-bound proteins were eluted from beads by the addition of 3× SDS loading buffer and boiling for 5 min for further test.

**Luciferase reporter assay.** Luciferase activity was measured using the dual-luciferase reporter assay system (Promega, E1910). Renilla luciferase expressed by pRL *Renilla* luciferase control reporter vectors (Promega, E2231) was used as an internal control to correct for differences in both transfection and harvest efficiency. To verify the target genes of miRNAs, cells were co-transfected with 50 nM miRNAs or NC duplex, 2 ng pRL *Renilla* luciferase control reporter vectors, and 20 ng firefly luciferase reporter plasmid that contained the wild type or mutant miRNA binding sequence of the 3′-UTR of *SRSF10* (GeneCopoeia, HmiT091024-MT06) or *TXNDC5* (GeneCopoeia, HmiT113146-MT06) for 48 h.

**Immunofluorescence.** For immunofluorescence, cells were seeded on coverslips in six-well plates and fixed with 4% paraformaldehyde for 20 min at room temperature. After three PBS-T (PBS with 0.1% tween-20) washes, cells were blocked with 5% BSA in PBS-T for 30 min at 37 °C, followed by a second blocking with Image-It FX Signal Enhancer Ready Probes Reagent (Thermo Fisher Scientific, R37107) 30 min at 37 °C. The cells were incubated with primary antibody (Supplementary Table 5) in PBS-T with 1% BSA overnight at 4 °C, after three wash (10 min each) and the slides were incubated with Alexa Fluor® 594 goat anti-rabbit (Molecular Probes, A-11037), Alexa Fluor® 488 goat anti-mouse (Molecular Probes, A-11029) or Alexa Fluor® 633 goat anti-chicken (Molecular Probes, A-21103) prepared in PBS-T with 2% BSA. The slides were counterstained with Hoechst 33342 (Molecular Probes, H3570) and imaged using the confocal laser-scanning microscope TCS SP8 STED (Leica, Weztlar, Germany).

**Immunohistochemistry (IHC) analysis.** Paraffin-embedded HCC tissue samples and tissues from mouse xenografts were cut into 5 μm sections and placed on poly-lysine-coated slides. The samples were deparaffinised in xylene and rehydrated by immersing samples in decreasing graded alcohol solutions. Heat-induced antigen retrieval was performed by heating in citrate buffer (pH 6.0) (Dako, S1699). Samples were blocked with 10% goat serum before incubation with the primary antibody. The samples were incubated overnight using the following primary antibodies: anti-SREK1^L (Customized made by Genscript E10; 1:3000), anti-SREK1 (Sigma-Aldrich, HPA037674; 1:100), anti-SRSF10 (Sigma-Aldrich HPA053805; 1:100), anti-Ki-67 (Sigma-Aldrich, SAB5300423; 1:100), anti-active Caspase 3 (CST, 9664S, 1:200) or an isotype-matched IgG as a negative control in a humidified container at 4 °C. IHC was performed with the Dako Envision plus System (Dako, GK500705) according to the manufacturer's instructions. The intensity of staining was evaluated on a scale of 1–4 according to the percentage of positive tumors. The protein expression of SREK1^L and SRSF10 was scored by two independent pathologists for the expression studies.

**Evaluation of the in vivo role of *SREK1^L* by hepatectomy-driven hepatocyte proliferation.** Liver regeneration after partial hepatectomy (PHx) may facilitate tumor growth and HCC recurrence. It remodels a microenvironment through alterations in cellular signaling pathways, which activate the proliferation of mature hepatocytes[60]. Thus, we utilized PHx to evaluate the effect of Srek1^L on hepatocyte proliferation. Gain-of-function experiments were performed through over-expressing *Srek1^L* in C57BL/6 mice by intravenous tail vein injection with control (AAV8-CMV-eGFP) or with *Srek1^L* overexpressing adeno-associated virus (AAV8-CMV-SREK1-IRES-eGFP) (Hanbio Biotechnology, assay ID: HH20210414HZCY-AAV01) with a titer 1 × 10^12 vg ml^−1. To evaluate the mRNA level of *Srek1^L*, mice were sacrificed 40 days post-AAV injection, and livers were isolated for GFP fluorescence detection and RNA extraction. GFP fluorescence detection was carried out by PhotoIMAGER OPTIMA (Biospace lab, Paris, France).). *Srek1^L* mRNA level is detected by real-time PCR. Mice livers were collected, frozen and embedded in OCT (Sakura 4583) compound 4th and 8th days post PHx. Livers blocks were sectioned and stained by phosphorylation of histone 3 serine 10 (anti-pH3S10; Abcam ab5176; 1:200) for proliferative potential evaluation.

**Percentage-splice-in (PSI), statistics and reproducibility.** AS of *SREK1* exon 10 in ten pairs tissue samples in Fig. 1a, b and Supplementary Fig. 5k (bottom panel) were represented by PSI, which was calculated by the number of expression (reads) where splice in occurred divided by the number of expression (reads) where both splice in and splice out occurred. The PSI value varied between 0 and 1, representing the percentage of reads where certain exons existed. All statistical analyses were performed using two-tailed Student's *t* test with either GraphPad Prism 7 version 1.0 (GraphPad Software, Inc., CA, USA) or the Partek® Genomics Suite® version 6.6 (Partek Incorporated Inc, MO, USA) unless otherwise indicated. Survival curves were calculated using the Log-rank (Mantel-Cox) test. Due to the big differential distribution of *SREK1^L*, *SREK1^S* and B-T expression in some tumor tissues, we have used the mean of gene expression to set the cutoff for survival analysis. However, due to the poor correlation of *SREK1^S* expression with the

survivals (Supplementary Fig. 1d) and the relatively big expression differences of $SREK1^L$ and $SREK1^S$ in some patients' tumor samples (Fig. 1b, e), we set the medium readings of $SREK1^L/SREK1^S$ as a cutoff to present better correlation with survival. Differences were considered statistically significant when $P$ values were less than 0.05. *$P < 0.05$, **$P < 0.01$ and ***$P < 0.001$ are used for all the analyses. At least three times experiments have been performed and repeated for the experiments in Figs. 1g, 2d, f, and 5e–g and Supplementary Figs. 2h, 3h, i, 4a, c, 5b (top panel), and 5c–f.

**Reporting summary**. Further information on experimental design is available in the Nature Research Reporting Summary linked to this article.

## Data availability

The RNA sequencing data generated in this study have been deposited in the GEO database under accession code GSE182102. The SILAC proteomics data of SRSF10 interactors generated in this study have been deposited in the PRIDE database under accession code PXD030800. The microarray data have been deposited in the ArrayExpress public database under accession codes E-MEXP-84 and E-TABM-292. The $SREK1^L$ binding sites on the B-T mRNA were predicted by RNA-Protein Interaction Prediction (RPISeq) online database [http://pridb.gdcb.iastate.edu/RPISeq/about.php]. The remaining data are available within the article, Supplementary Information, or Source Data file. Source Data are provided with this paper.

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

## Acknowledgements

We thank the SingHealth Tissue Repository (STR) for providing the tissues, IHC staining and scoring, and SingHealth Advanced Bioimaging Core for assisting in confocal imaging. This work was funded by the National Natural Science Foundation of China (81802338 and 82072646), Zhejiang Provincial Natural Science Foundation of China (LR21H160001), "Pioneer" and "Leading Goose" R&D Program of Zhejiang Province (2021C03G2153004), and Start-up Grant of HZNU (4125C5021820470) (to J.C.); National Medical Research Council of Singapore (to K.M.H.); National Natural Science Foundation of China (81973635 and 81730108) (to T.X.); National Natural Science Foundation of China (81802831) (to C.C.); National Natural Science Foundation of China (81903143) (to Y.Q.), Operating grant MOP-136948 from Canadian Institutes of Health Research (to B.C.) and Singapore Ministry of Health's National Medical Research Council Open Fund Young Individual Research Grant (MOH-OFYIRG19Nov-0016)

(to Y.C.). Images in Supplementary Fig. 2k sourced from Servier Medical Art at smart.servier.com.

## Author contributions

J.C., T.X., and K.M.H provided the study concept and design. C.J.C., M.R., Y.T.Q., H.D., G.X.C., and J.C. interpreted and analyzed the data. J.C., C.J.C., M.R., Y.T.Q., H.D., Y.W., H.X., A.D., M.J.W., Heng.G., M.Q.S., Y.Q.N., Q.L., Z.Y., M.L.W., S.S.C., S.J.Z., D.J.C., L.N.L., F.Y., Hao.G., B.D.C., L.S., L.X., Y.C., G.X.C., and J.G. performed the experiments. A.D., J.P.S., and G.Y.C. did the mice work and collected the tissues' samples and the clinical parameter data. L.T., K.S., and V.P.S. performed the bioinformatics analysis. D.S.G and B.C. developed and prepared the SRSF10 inhibitors. J.C. and K.M.H. wrote the manuscript. W.J.H. and C.J.C. revised the manuscript. All authors approved the final version of the manuscript.

## Competing interests

The authors declare no competing interests.
