## [Peer Review File · Nature Communications]

The aberrant upregulation of exon 10-inclusive SREK1 through SRSF10 acts as an oncogenic driver in human hepatocellular carcinomaREVIEWER COMMENTS

Reviewer #1, expert in HCC and models (Remarks to the Author):

This study revealed a novel SRSF10/SREK1/BLOC1S5-TXNDC5 oncogenic signalling loop in hepatocellular carcinoma (HCC). Exon 10-inclusive SREK1 (SREK1L) highly expressed in HCC tissues, which sustain the expression of BLOC1S5-TXNDC5 transcript. BLOC1S5-TXNDC5 as a ceRNA promotes SRSF10 and TXNDC5 expression by inhibiting miR-30c-5p and miR-30e-5p, and splicing factor SRSF10 could sustain the level of SREK1L. The study data support the conclusions. However, minor issues should be addressed.

1. The key role of TXNDC5 in the regulation of tumor growth by SREK1L needs to be confirmed.
2. How to quantify the relative expression of the variants, SREK1L/SREK1S, in Figure 1C? The method of normalization in Figure 1F may be more appropriate.
3. In Figure 1E, HCC patients with high and low expression of SREK1L were divided using the mean as the cut-off value, but patients in Figure 1G were grouped by the median value of SREK1L/SREK1S ratio, and Figure 1I used 2 as the cut-off value. Why? The cut-off values were not indicated in Figure 3D and S3D.
4. Immunohistochemical staining should be performed to confirm the knockdown of SREK1L in xenograft tumors in Figure 2D as well as the knockdown of SRSF10 in xenograft tumors in Figure 6F/G.
5. Whether the TCGA database supports the conclusions that SREK1L maintains the expression of B-T or that SRSF10 promotes SREK1L expression in HCC?
6. The nuclear and cytoplasmic SREK1 i in siSRSF10- or siScram-transfected HCC cells can analyzed by western blotting after nuclear-cytosol extraction.
7. The protein levels of SREK1L and SREK1S in HCC cell lines were analyzed by western blotting in Figure 5G and 7D. Are there antibodies that can distinguish these two isomers?
8. More discussion about the oncogenic mechanisms of TXNDC5 and SRSF10 should be supplemented, especially their roles in tumor growth.
9. The full name of the abbreviation HCC-T that appears for the first time in the Page 5 Line 6 should be noted.
10. Page 7 Line 4, "an tissue set" should be corrected to "a tissue set".
11. Page 17 Line1 "(Fig.S6F)" is missing a space.

Reviewer #2, expert in RNA metabolism (noncoding biology and NMD decay) (Remarks to the Author):

In this study the authors focus on the regulation of exon 10 in the SREK1 (Splicing Regulatory Glutamic Acid and Lysine Rich Protein) pre-mRNA during human hepatocellular carcinoma.

First, the authors observed an increased inclusion of exon 10 (SREK1L isoform) in HCC tissues and this elevated expression of the exon 10-containing isoform correlated well with poor prognosis. Next, the authors show that SREK1L isoform promotes hepatocarcinogenesis. For this, they used siRNAs/shRNAs specifically targeting exon 10 to deplete the SREK1L isoform in Hep3B and

HCCLM3 cells, as well as in mouse tumorigenic assays.

They went on to show that SREK1L promotes the expression of BLOC1S5-TXNDC5 (B-T), a noncoding targeted gene of nonsense-mediated mRNA decay (NMD) via inhibiting the exon-exon junction complex (EJC) binding and subsequent decay signals. It is not entirely clear though why the authors focused almost exclusively on B-T expression. Next, they show that B-T further acts as a downstream effector of SREK1L via its competing endogenous RNA (ceRNA) leading to an inhibition of two miRNAs, miR-30c-5p and miR-30e-5p and promoting the downstream oncogenic targets SRSF10 and TXNDC5.

The paper next takes yet another turn and focuses on the role of the SR protein SRSF10 in promoting E10 inclusion in the SREK1 pre-mRNA. They propose that this SRSF10 AS-regulated event acts as an oncogenic driver in human hepatocellular carcinoma.

There are some interesting aspects to this study, but overall it is in a preliminary stage. The narrative is confusing in parts and sometimes too many experiments or lines of research are not better than a focused project.

There are some parts of this paper that could make a reasonable story, mostly focusing on the role of SREK1L isoform in HCC and its regulation by SRSF10. By contrast the sections focused on the role of SREK1 in the inhibition of B-T NMD-mediated degradation as well as the role of B-T as a ceRNA are far less convincing.

Specific comments

- It remains a bit of a mystery what prompted the authors to look for SREK1 AS regulation, and in particular, inclusion/skipping of exon 10 in the context of HCC. This could be added to the Introduction or at the beginning of the Results section
- On Figure 2, the authors should include as a control siRNAs/shRNAs that target both isoforms of SREK1 or preferably one that targets the junction of Exon 9-11 to be able to compare depletion of the E10-containing with the E10-skipped isoform.
- On Figure 3, the authors show that knockdown of SREK1-exon 10 containing isoform leads to an upregulation of the BT ncRNA. A control to show that this is exclusively due to the specific knockdown of the E10-containing isoform and not to SREK1 pre-mRNA (both S and L) seems to be missing
- On page 11, the authors claim that NMD is strictly controlled by two protein complexes, the EJC and the SURF complex. This is somehow inaccurate. Whereas EJC is an enhancer of NMD, there are many cases of NMD targets that are regulated independently of an EJC, such as those harboring a long 3'UTR
- On Fig. 3F, it is not clear at all that the interactions with UPF1 and MOV10 are specific for the SREK1L isoform. Was the antibody used specific for the L isoform? Where is the experiment using an antibody that recognizes both the short and Long isoforms?
- There is no direct proof that SREK1L (or the S isoform) bind to the exon-exon junction in the BT transcript and protect it from EJC binding. There could be alternative mechanisms by which SREK1L knockdown leads to an enhanced binding of Magoh or other EJC components. To conclusively demonstrate this, the authors should carry out EJC deposition assays.
- Another way of proving a role for SREK1L in NMD, would be to analyze its role in the absence of UPF1. What would be the outcome of an increased expression of SREK1L in the absence of UPF1? If HCC is triggered by the action of SREK1L on the NMD-mediated degradation of B-T, then the absence of UPF1 would make the levels of SREK1L irrelevant

Reviewer #3, expert in molecular mechanisms of alternative splicing (Remarks to the Author):

Summary:

In the present manuscript, the authors provide evidence that a splice isoform of SREK1 in which exon 10 is included (SREK1L) is upregulated and may be an oncogenic driver of hepatocellular carcinoma. They provide a series of in vivo functional assays and suggest some of the mechanisms involved, including the involvement of SRSF10 in the regulation of SREK1 exon 10 inclusion.

Major comments:

1. In regards to the standard PCR data presented in Figure 1C from 10 patients, this data is not entirely convincing. Repeats are needed for each so that the band intensities can be quantified statistically. This is especially critical as the number of patients is small.
2. The antibody specifically against SREK1L is a very useful resource. It would be necessary however to show that this antibody is specific through other means, e.g. western blot. Also, this antibody should be used to show that SREK1 isoform ratios are altered in a substantial cohort of patients at a protein level, ideally in the same patients examined by RTPCR in panel 1C.
3. Figure 2. These data are convincing. But perhaps to draw the conclusion that SREK1L is an oncogenic driver in itself is a premature conclusion - after all, this data is obtained will cell lines. A full in vivo model, eg a transgenic mouse model would be required to show conclusively that SREK1L is an oncogenic driver.
4. The striking difference in rMATS results from the two cell lines underlines the limitations of cell line models. However the observation that SREK1 might be involved in the regulation of NMD pathways is interesting, and it is not atypical to find "splice factors" involved in multiple posttranscriptional processes. However the choice to focus on one NMD-associated transcript here, seems a little arbitrary. So in order to establish more firmly that SREK1 is involved in tumorigenesis the modulation of NMD in HCC would require some additional evidence, e.g. at the very least, the association with other NMD targets.
5. In a similar vein, it seems clear that SRSF10 is involved in the regulation of SREK1 exon 10 inclusion, but the model presented in Figure 7I is perhaps too speculative at present - whether or not all these proteins are involved in the regulation of SREK1 exon 10 inclusion would require additional experiments. More generally, the claim of a "tumorigenic axis" is perhaps speculative, and a product of the history of the experiments conducted, rather than necessarily a genuine causative axis in HCC.

Minor comments:

1. Figure 1C, as well as a potential change in the rate of inclusion of exon 10, is there an impression here that SREK1 is upregulated in the tumours too on the whole? worth clarifying. Also, it is conventional in the splicing field, when describing a cassette exon, to express its inclusion as a "PSI" number and this should be a value between 0 and 1.
2. Clarify the presence of a doublet in the SRSF10 western blot, Fig. 5G and 7G.
3. Figure 5G, were these western blots repeated, and could they be quantified?
4. Figure 5E, and additional control would be to show that for example. the SREK1 exon 10 RNA does not co-IP with a non-involved splice factor such as SRSF1?

We thank all the reviewers and editor for their thoughtful comments and constructive suggestions to improve our manuscript. We have made point-by-point response according to the suggestions of the reviewers and editor.

Point-by-point response:

Response to Reviewer #1:

Question (Q) 1: The key role of TXNDC5 in the regulation of tumor growth by SREK1L needs to be confirmed.

Response (R) 1: To verify whether TXNDC5 mediates the role of SREK1^L on tumor growth promotion, we re-expressed the TXNDC5 in SREK1^L or scramble control knockdown HCCLM3 cells (*new Supplementary Figure 4C*) and found that expression of TXNDC5 could promote the cell growth and partially rescue the knockdown effect of SREK1^L on the cell growth inhibition in HCCLM3 cells (*new Supplementary Figure 4D*).

Q2: How to quantify the relative expression of the variants, SREK1L/SREK1S, in Figure 1C? The method of normalization in Figure 1F may be more appropriate.

R2: As the reviewer suggested, together with the question 1 of the reviewer 3, we have re-quantified the relative expression of the SREK1^L and SREK1^S variants in the ten pairs of HCC tissues samples. Further, we use the percent spliced in (PSI) to accurately evaluate the SREK1 splicing in tissues, and these data were presented in *new Figure 1A and 1B*.

Q3: In Figure 1E, HCC patients with high and low expression of SREK1L were divided using the mean as the cut-off value, but patients in Figure 1G were grouped by the median value of SREK1L/SREK1S ratio, and Figure 1I used 2 as the cut-off value. Why? The cut-off values were not indicated in Figure 3D and S3D.

R3: Due to the big differential distribution of SREK1^L, SREK1^S and B-T expression in some tumor tissues, we have used the mean of gene expression to set cutoff for all the survival analysis. Our data indicated that high expression of SREK1^L, but not SREK1^S, is significantly correlated with poor patients' OS and DFS (*new Figure 1D and Supplementary Figure 1D*). However, due to the poor correlation of SREK1^S expression with the survivals (*new Supplementary Figure 1D*) and the relatively big expression differences of SREK1^L and SREK1^S in some patients' tumor samples (*new Figure 1B and 1E*), setting the mean readings of SREK1^L/SREK1^S as cutoff thus could not show significant correlation ($P=0.1419$) with OS (*Figure R1A*). We then have to set the medium or 25/75 percentile of SREK1^L/SREK1^S for cutoff. Our analysis indicated that high SREK1^L/SREK1^S is significantly correlated with poor OS when using the medium (*new Figure 1F*) or 25/75 percentile (*Figure R1B*) as cutoff, and cutoff by medium ($P=0.0060$) show better correlation than by 25/75 percentile ($P=0.0354$). Thus, we finally used the medium as the cutoff for correlation analysis of SREK1^L/SREK1^S with survival in the manuscript.

Figure R1 Setting the mean or 25/75 percentile of SREK1^L/SREK1^S as cutoff for OS analysis.

As the reviewer suggested, for consistency, we used the mean of readings as cutoff for the correlation analysis of the survival with SREK1^L (*new Figure 1D*), SREK1^S (*new Supplementary Figure 1D*), B-T (*new Figure 3D*), A-A and S-D (*new Supplementary Figure 3D*), which were indicated in the corresponding figures legends. Particularly, we declared the rationale of using medium as cutoff for the correlation analysis of survival with SREK1^L/SREK1^S in *revised Material and Methods*.

Q4: Immunohistochemical staining should be performed to confirm the knockdown of SREK1L in xenograft tumors in Figure 2D as well as the knockdown of SRSF10 in xenograft tumors in Figure 6F/G.

R4: As the reviewer suggested, we have detected the SREK1^L and SRSF10 protein expression in the xenograft tumors by immunohistochemical staining, which were shown in *new Figure 2F and 6G*.

Q5: Whether the TCGA database supports the conclusions that SREK1L maintains the expression of B-T or that SRSF10 promotes SREK1L expression in HCC?

R5: To answer the question raised by the reviewer, we analyzed the *Pearson* correlation analysis for the gene expression of *SREK1*, *B-T* and *SRSF10* in TCGA-LIHC gene expression database. It was found that *SREK1* is not significantly correlated with *B-T* in TCGA-HCC tissues (*Figure R2*), which might be due to the relatively low expression and sample-dependent regulation of NMD signals and/or targeted genes in HCC tissues. However, we still found that *SREK1* expression was very significantly correlated with *SRSF10* expression in TCGA-LIHC database (Pearson correlation, $r=0.74$, $P=0$), which partially supports our conclusion SRSF10 promoting SREK1 expression, and we included this data in *new Supplementary Figure 5J*.

Figure R2 Correlation analysis of *SREK1* and *B-T* expression in TCGA tumor tissues via GEPIA

(<http://gepia.cancer-pku.cn/index.html>).

Q6: The nuclear and cytoplasmic SREK1 in siSRSF10- or siScram-transfected HCC cells can analyzed by western blotting after nuclear-cytosol extraction.

R6: As the reviewer suggested, we have performed the nuclear and cytoplasmic extraction for SREK1 splicing variants in siSRSF10 or siScram transfected HCCLM3 cells. Our data suggested that SRSF10 knockdown could induce the production of the SREK1^S accumulated in cytoplasm, but not in nucleus. The data was presented in *new Supplementary Figure 5F*.

Q7: The protein levels of SREK1L and SREK1S in HCC cell lines were analyzed by western blotting in Figure 5G and 7D. Are there antibodies that can distinguish these two isomers?

R7: Together with the other two reviewers' questions, we have tested all the commercial antibodies of SREK1 to investigate which could recognize both forms of long and short SREK1. By knockdown of SREK1^S and/or SREK1^L with siRNA (*Supplementary Figure 2G and 2H*), we found one SREK1 antibody (from Thermo Fisher) could recognize both SREK1^S and SREK1^L (*Supplementary Figure 2H*), which was used for further revised experiments.

Q8: More discussion about the oncogenic mechanisms of TXNDC5 and SRSF10 should be supplemented, especially their roles in tumor growth.

R8: As suggested by the reviewer, we have made further discussion on the recent updates of the potential oncogenic mechanisms of TXNDC5 and SRSF10 in cancers, which was discussed in the *new discussion part*.

Q9: The full name of the abbreviation HCC-T that appears for the first time in the Page 5 Line 6 should be noted.

R9: We have revised the abbreviation as the reviewer mentioned.

Q10: Page 7 Line 4, "an tissue set" should be corrected to "a tissue set".

R10: We have corrected the word as the reviewer suggested.

Q11: Page 17 Line1 "(Fig. S6F)" is missing a space.

R11: We have added a space as the reviewer suggested.

Response to Reviewer #2:

Q1: It remains a bit of a mystery what prompted the authors to look for SREK1 AS regulation, and in particular, inclusion/skipping of exon 10 in the context of HCC. This could be added to the Introduction or at the beginning of the Results section

R1: As the reviewer suggested, we have added more introduction for the purpose on studying SREK1 splicing in the *new introduction part*.

Q2: On Figure 2, the authors should include as a control siRNAs/shRNAs that target both isoforms of SREK1 or preferably one that targets the junction of Exon 9-11 to be able to compare depletion of the E10-containing with the E10-skipped isoform.

R2: As the reviewer suggested, we designed three siRNA sequences crossing the exon9-11 junction targeting SREK1^S. The knockdown experiments showed that the siSREK1^S could silence the SREK1^S specifically and have no effect on the expression of SREK1^L protein (*new Supplementary Figure 2G and 2H*). The cell growth analysis indicated that knockdown of SREK1^S have no obvious inhibition or promotion on the cell growth in Hep3B cells (*new Supplementary Figure 2I*).

Q3: On Figure 3, the authors show that knockdown of SREK1-exon 10 containing isoform leads to an upregulation of the BT ncRNA. A control to show that this is exclusively due to the specific knockdown of the E10-containing isoform and not to SREK1 pre-mRNA (both S and L) seems to be missing

R3: As the reviewer suggested, we have silenced L or S isoforms of SREK1 and detected the B-T expression by PCR. Our data indicated that knockdown of the SREK1^L, but not the SREK1^S, could lead to the significant upregulation of the B-T expression (*new Supplementary Figure 3F*).

Q4: On page 11, the authors claim that NMD is strictly controlled by two protein complexes, the EJC and the SURF complex. This is somehow inaccurate. Whereas EJC is an enhancer of NMD, there are many cases of NMD targets that are regulated independently of an EJC, such as those harboring a long 3'UTR

R4: We aware the inaccurate description on the NMD regulation as the reviewer mentioned. Thus, we have corrected the description in the *new results section*.

Q5: On Fig. 3F, it is not clear at all that the interactions with UPF1 and MOV10 are specific for the SREK1L isoform. Was the antibody used specific for the L isoform? Where is the experiment using an antibody that recognizes both the short and Long isoforms?

R5: Yes, the antibody 1 (ab1) used in pull-down experiment in Figure 3F could recognize and pull down the SREK1^L, and the ab2 could recognize and pull down both the SREK1^L and SREK1^S, as indicated in our data (*new Figure 3E*). This data indicated that SREK1^L could interact with UPF1 and MOV10 complexes. To investigate whether SREK1^S also could interact with UPF1 and MOV10 complexes, we performed a pull-down experiment using Flag tagged SREK1^S or SREK1^L overexpression in Hep3B cells and found that transiently expressed SREK1^S also could interact with UPF1 and MOV10 compared with SREK1^L (*Figure R3*). However, due to the cytoplasmic accumulation property of SREK1^S, relatively weaker interaction with UPF2 and MAGOH (nuclear EJC components) has been detected in Flag-SREK1^S expressed cells, compared with

Flag-SREK1^L expressed cells (*Figure R3*). This indicates that the interaction of SREK1 variant proteins with SURF and EJC complex is not dependent on the EK domain which is important for nuclear SREK1 regulatory roles. As we confirmed previously, most of the endogenous SREK1 is the long form (SREK^L), we think the interaction with SURF and EJC complex is mainly mediated by SREK1^L in HCC cells.

Figure R3 The immunoprecipitation analysis of the interaction of Flag-tagged SREK^L or SREK1^S with SURF and EJC complexes proteins in Hep3B cells.

Q6: There is no direct proof that SREK1L (or the S isoform) bind to the exon-exon junction in the BT transcript and protect it from EJC binding. There could be alternative mechanisms by which SREK1L knockdown leads to an enhanced binding of Magoh or other EJC components. To conclusively demonstrate this, the authors should carry out EJC deposition assays.

R6: We appreciate the suggestion by the reviewer, and further performed the EJC deposition assays as the reviewer suggested. Flag-tagged SREK1^L and endogenous EJC components MAGOH and UPF2 were coprecipitated with biotinylated B-T or B-T ΔBS (SREK1^L binding site deleted) mRNA in Hep3B nuclear extract. We found that increased Flag-SREK1^L expression could inhibit the deposition of UPF2 and MAGOH, two EJC components, with the B-T, but not with B-T ΔBS RNA (*Fig. S3I*). Our data indicated that SREK1^L is involved in regulating the EJC deposition with B-T in HCC cells.

Q7: Another way of proving a role for SREK1L in NMD, would be to analyze its role in the absence of UPF1. What would be the outcome of an increased expression of SREK1L in the absence of UPF1? If HCC is triggered by the action of SREK1L on the NMD-mediated degradation of B-T, then the absence of UPF1 would make the levels of SREK1L irrelevant

R7: As the reviewer suggested, we did knockdown of UPF1 and scramble control in shScram and shE10#1 Hep3B and HCCLM3 cells (*new Supplementary Figure 3L*) and the BrdU proliferation analysis was performed further to evaluate the cell growth potential. Our data showed that knockdown of SREK1^L could still inhibit the cell proliferation (*new Supplementary Figure 3L*). However, when we depleted the endogenous UPF1 by siRNA in the cells, knockdown of SREK1^L failed to inhibit the cell proliferation further (*new Supplementary Figure 3L*), indicating that the cell growth promotion role of SREK1^L in HCC cells could partially depend on the present and regulatory role of UPF1, as the reviewer predicted.

Response to Reviewer #3:

Q1: In regards to the standard PCR data presented in Figure 1C from 10 patients, this data is not entirely convincing. Repeats are needed for each so that the band intensities can be quantified statistically. This is especially critical as the number of patients is small.

R1: Together with the question 6 (Q6), as the reviewer suggested, we have repeated the experiments and quantified the bands for PSI to indicate the relative spliced level of SREK1^L and SREK1^S variants. These data were presented in *new Figure 1A and 1B*.

Q2: The antibody specifically against SREK1L is a very useful resource. It would be necessary however to show that this antibody is specific through other means, e.g. western blot. Also, this antibody should be used to show that SREK1 isoform ratios are altered in a substantial cohort of patients at a protein level, ideally in the same patients examined by RTPCR in panel 1C.

R2: As the reviewer suggested, we have tested all the commercial antibodies of SREK1 to investigate which could recognize both forms of long and short SREK1. By knockdown of SREK1^S or SREK1^L with siRNA, we found one SREK1 antibody (from Thermo Fisher) could recognize both SREK1^S and SREK1^L (*Supplementary Figure 2G and 2H*), which was used for further experiments. We also found that knockdown of SREK1^S has no significant effect on the growth rate of Hep3B cells, compared with scramble control or SREK1^L knockdown (*Supplementary Figure 2I*).

Moreover, as the reviewer suggested, we further tested SREK1^S and SREK1^L protein expression in six pair tissue samples used in new Figure 1A by western blot. We found significant higher PSI for SREK1^L and SREK1^S splicing in HCC than in match normal tissues (*Supplementary Figure 5K*), which is consistent with our gene expression data.

Q3: Figure 2. These data are convincing. But perhaps to draw the conclusion that SREK1L is an oncogenic driver in itself is a premature conclusion - after all, this data is obtained will cell lines. A full in vivo model, eg a transgenic mouse model would be required to show conclusively that SREK1L is an oncogenic driver.

R3: We appreciate the recommendation on the *in vivo* model to investigate the proliferative role of SREK1^L in hepatocyte. Indeed, we planned to construct a transgenic mouse, but due to the Covid-19, there was some delay during the process and the construction is still on-going. To fast evaluate the *in vivo* proliferative role of SREK1^L in hepatocytes, we employed a liver regeneration model by resection after adenovirus-mediated gene delivery and expression in liver (*Supplementary Figure 2K-L*). Our data showed that forced expression of SREK1^L in liver could significantly improve the cell division rate (marked by phosphorylated Histone 3 Ser10, pH3S10) to accelerate the proliferation of hepatocyte compared with the GFP vector control on the day 4th and 8th post the liver resection (*Supplementary Figure 2M and 2N*), further confirming a potential driver role of SREK1^L on hepatocytes' proliferation *in vivo*.

Q4: The striking difference in rMATS results from the two cell lines underlines the limitations of cell line models. However the observation that SREK1 might be involved in the regulation of NMD pathways is interesting, and it is not atypical to find "splice factors" involved in multiple posttranscriptional processes. However the choice to focus on one NMD-associated transcript here, seems a little arbitrary. So in order to establish more firmly that SREK1 is involved in

tumorigenesis the modulation of NMD in HCC would require some additional evidence, e.g. at the very least, the association with other NMD targets.

R4: As the reviewer suggested, we have tested the function of other two potential SREK1^L-regulated NMD targeted genes – A-A and S-D by siRNA-mediated knockdown in Hep3B and HCCLM3 cells (*Figure R4A-B*). Our data indicated that knockdown of A-A and S-D has no significant inhibition or promotion on the cell proliferation and also failed to rescue the knockdown effect on the proliferation by SREK1^L siRNA treatment (*Figure R4C*). These data indicate that B-T is the main NMD targeted gene to mediate SREK1^L's growth promotion effect. Thus, we did not put this data in main text.

Figure R4 The realtime PCR analysis of the A-A and S-D expression after **A)** SREK1^L or **B)** A-A/S-D knockdown in Hep3B and HCCLM3 cells. **C)** BrdU proliferation analysis of the knockdown of each targeted genes in Hep3B and HCCLM3 cells.

Q5: *In a similar vein, it seems clear that SRSF10 is involved in the regulation of SREK1 exon 10 inclusion, but the model presented in Figure 7I is perhaps too speculative at present - whether or not all these proteins are involved in the regulation of SREK1 exon 10 inclusion would require additional experiments. More generally, the claim of a "tumorigenic axis" is perhaps speculative, and a product of the history of the experiments conducted, rather than necessarily a genuine causative axis in HCC.*

R5: We agree with the comment of the reviewer on the model presented in Figure 7I. As we have not generated enough data to verify all these SRSF10 interactors involved in regulating the SREK1 exon10 splicing, we thus deleted the model proposed in Figure 7I. Moreover, as the reviewer suggested, we revised all the claim of "axis" in the text.

Q6: *Figure 1C, as well as a potential change in the rate of inclusion of exon 10, is there an impression here that SREK1 is upregulated in the tumours too on the whole? worth clarifying. Also, it is conventional in the splicing field, when describing a cassette exon, to express its inclusion as a "PSI" number and this should be a value between 0 and 1.*

R6: As the reviewer suggested, we have revised the old Figure 1C by re-calculating the PSI for

each samples to evaluate the splicing of exon10. This new data with PSI value has been shown in new Figure 1A and 1B.

Q7: *Clarify the presence of a doublet in the SRSF10 western blot, Fig. 5G and 7G.*

R7: Previously, we have also noticed the doublet in the SRSF10 immunoblotting detection. Based on the previous reports on the SRSF10 inhibitor – 1C8, we speculated that it might be the phosphorylated SRSF10 in HCC cell lines. To confirm it, we treated the HCCLM3 (most of the SRSF10 is modified in our WB data) cells' lysate with λ -phosphatase (λ -PPase) and found that the upper band was disappeared in the phosphatase treated sample compared with the untreated group (Figure R5). Our data confirm that the upper bands of SRSF10 immunoblotting are indeed the phosphorylated SRSF10.

Figure R5 The immunoblotting analysis of the SRSF10 bands in λ -phosphatase treated or untreated HCCLM3 cell lysate.

Q8: *Figure 5G, were these western blots repeated, and could they be quantified?*

R8: Yes, we had repeated these western blots and did the quantification by Image J software. The quantified data were used further for the correlation analysis in Figure 5H and Supplementary Figure 5G. Moreover, we also analyzed the correlation of protein expression of SREK1^S and SRSF10, which were indicated that SREK1^S is not very well correlated ($r=0.4287$) with SRSF10 expression (Figure R6) below.

Figure R6 The correlation analysis of protein expression of SREK1^S and SRSF10 in HCC cell lines.

Q9: *Figure 5E, and additional control would be to show that for example, the SREK1 exon 10 RNA does not co-IP with a non-involved splice factor such as SRSF1?*

R9: As the reviewer suggested, we have checked whether SRSF1 could co-IP with SREK1 exon10 RNA in Hep3B and HCCLM3 cells. Our data indicated that SRSF10, but not SRSF1, could co-IP with SREK1 exon10 RNA (new Supplementary Figure 5D).

REVIEWERS' COMMENTS

Reviewer #1 (Remarks to the Author):

The authors have addressed most of the reviewer's questions and the manuscript is prominently improved.

A minor issue that should be addressed:

The order of Figure 2D and 2E needs to be changed.

Reviewer #3 (Remarks to the Author):

I would like to thank the authors for addressing my comments fully. I would maintain that a transgenic mouse model would have been ideal, but I fully appreciate the practicalities of getting it done on time. The addition of the liver regeneration model does at least strengthen the in vivo significance of the work.

Reviewer #4 (Remarks to the Author):

The authors have adequately responded to Reviewer #2's previous comments.

We thank all the reviewers and editors for their positive comments and constructive suggestions to improve our manuscript. We have made point-by-point response according to the suggestions of the reviewers and editor.

Point-by-point response:

Response to Reviewer #1:

Question: The authors have addressed most of the reviewer's questions and the manuscript is prominently improved. A minor issue that should be addressed: The order of Figure 2D and 2E needs to be changed.

Response: We appreciate for the detailed suggestion and we have revised the order of Figure 2D and 2E as suggested by the reviewer, which have been organized in new Figure 2.

Response to Reviewer #3:

Q1: I would like to thank the authors for addressing my comments fully. I would maintain that a transgenic mouse model would have been ideal, but I fully appreciate the practicalities of getting it done on time. The addition of the liver regeneration model does at least strengthen the in vivo significance of the work.

R1: We fully agree with and understand the suggestion of the reviewer, and appreciate the positive final comment from the reviewer.

Response to Reviewer #4:

Q1: The authors have adequately responded to Reviewer #2's previous comments.

R1: We appreciate the reviewer for the positive comment on our revised manuscript.